

# Trends and abrupt changes in 104-years of ice cover and water temperature in a dimictic lake in response to air temperature, wind speed, and water clarity drivers

M. R. Magee[1], C. H. Wu[1], D. M. Robertson[2], R. C. Lathrop[3], and D. P. Hamilton[4]

[1]{Civil and Environmental Engineering, University of Wisconsin-Madison, Madison, Wisconsin, USA}

[2]{Wisconsin Water Science Center, U.S. Geological Survey, Middleton, Wisconsin, USA}

[3]{Center for Limnology, University of Wisconsin-Madison, Madison, Wisconsin, USA}

[4]{Environmental Research Institute, University of Waikato, Hamilton, New Zealand}

Correspondence to: C.H. Wu (chinwu@engr.wisc.edu)

**Abstract**

The one-dimensional hydrodynamic-ice model, DYRESM-WQ-I, was modified to simulate ice cover and thermal structure of dimictic Lake Mendota, WI, USA, over a continuous 104-year period (1911-2014). The model results were then used to examine the drivers of changes in ice cover and water temperature, focusing on the responses to shifts in air temperature, wind speed, and water clarity at multi-year time scales. Observations of the drivers include a change in the trend of warming air temperatures from 0.081 °C per decade before 1981 to 0.334 °C per decade thereafter, as well as a shift in mean wind speed from 4.44 m s$^{-1}$ to 3.74 m s$^{-1}$ in 1994. Observations show that Lake Mendota has experienced significant changes in ice cover: later ice on (9.0 days later per century), earlier ice-off (12.3 days per century), decreasing ice cover duration (21.3 days per century), while model simulations indicate a change in maximum ice thickness (12.7 cm decrease per century). Model simulations also show changes in the lake thermal regime of: earlier stratification onset (12.3 days per century), later fall turnover (14.6 days per century), longer stratification duration (26.8 days per century), and decreasing summer hypolimnetic temperatures (-1.4 °C per century). Correlation analysis of lake variables and driving variables revealed ice cover variables,





stratification onset, epilimnetic temperature, and hypolimnetic temperature were most closely
correlated with air temperature, whereas freeze-over water temperature, hypolimnetic heating,
and fall turnover date were more closely correlated with wind speed. Each lake variable (i.e.,
ice-on and ice-off dates, ice cover duration, maximum ice thickness, freeze-over water
temperature, stratification onset, fall turnover date, stratification duration, epilimnion
temperature, hypolimnion temperature, and hypolimnetic heating) was averaged for the three
periods (1911-1980, 1981-1993 and 1994-2014) delineated by abrupt changes in air
temperature and wind speed. Average summer hypolimnetic temperature and fall turnover
date exhibit significant differences between the third period and the first two periods.
Changes in ice cover (ice-on and ice-off dates, ice cover duration, and maximum ice
thickness) exhibit an abrubt change after 1994which was related in part to the warm El Niño
winter of 1997−1998. Under-ice water temperature, freeze-over water temperature,
hypolimnetic temperature, fall turnover date, and stratification duration demonstrate a
significant difference in the third period (1994−2014), when air temperature was warmest and
wind speeds decreased rather abruptly. The trends in ice cover and water temperature
demonstrate responses to both long-term and abrupt changes in meteorological conditions that
can be complemented with numerical modelling to better understand how these variables will
respond in a future climate.

## 20 1 Introduction

Many studies have shown that lake temperatures and ice cover can strongly affect water
chemistry, individual organism physiology, population abundance, community structure, and
food-web dynamics (Weyhenmeyer et al. 1999; Straile, 2000; Gerten and Adrian 2000;
Arhonditsis et al. 2004b). Air temperature, wind speed, and water clarity are important factors
driving these lake ecosystem properties. Understanding how lakes respond to changes in these
drivers is of great interest to predict how lakes may change in the future (Robertson 1989;
Magnuson et al. 1997; Fang and Stefan 2009). The long-term response of lake ice and water
temperature to changing air temperature and wind speed is integral to assessment of the
potential impacts of climate change on water quality and ecology of lakes.
Over the past 100 years, climate has been changing and will continue to change (IPCC 2013).
Globally-averaged combined land and ocean surface temperature data show a linear warming
trend of 0.85 ℃ from 1880−2012 (IPCC 2013). This warming was most pronounced from





1979−2012, greater than 0.25 ℃ per decade (Hartmann et al., 2013) Increases in air
temperature alter the ice cover of lakes (Robertson et al. 1992; Magnuson et al. 2000; Butcher
et al. 2015) and affect their thermal structures (Robertson and Ragotzkie 1990), evidenced by
increasing epilimnetic temperatures (Schindler et al. 1990; Arhonditsis et al. 2004a; Dobiesz
and Lester 2009), warming of the lake surface temperature (Schneider and Hook, 2010;
Shimoda et al. 2011), increasing temperature gradient across the thermocline (King et al.
1997; Livingstone 2003), changing thermocline depth (Schindler et al. 1990; King et al.
1997), advancing the onset of summer stratification (Austin and Colman 2007), delaying fall
turnover (King et al. 1997), increasing the strength of thermal stratification (Rempfer et al,
2010), and prolonging the stratified period (Robertson and Ragotzkie 1990; Wilhelm and
Adrian 2008).
Trends in wind speed over the last 30–50 years have been reported in several studies that have
analyzed historical wind speed records across the globe (Jiang et al. 2010; Wan et al. 2010).
Klink (2002) examined 22- to 35-year records (ranging between 1959–1995) of wind speed at
seven stations in and around Minnesota and found decreasing annual wind speeds at five of
the seven stations. Pryor et al. (2009) reported that the 50[th] and 90[th] percentile annual wind
speeds over the period 1973–2005 across most of the U.S. have also decreased. Decreased
wind speeds increase thermal stratification and can reduce whole-lake average temperature
(Tanentzap et al. 2008). Interestingly, an opposing trend (increasing wind speed) has been
observed in Lake Superior, North America, where the lake surface temperatures have been
warming faster than air temperatures (Austin and Colman 2007). Desai et al. (2009) suggest
that the larger increase in water temperatures than air temperatures reduced the air-water
temperature gradient and destabilized the atmospheric surface layer above Lake Superior,
which resulted in  increasing wind speed at a rate of nearly 5% per decade. Differences in
wind-driven mixing may explain different temperature responses of hypolimnetic waters in
large and small lakes (Winslow et al. 2015). While the importance of wind in lake heat
transfer (Fu et al. 2009; Read et al. 2012), mixing, and thermal structure (Schindler et al.
1990; Desai et al. 2009) has been recognized, studies on the effects of wind speed alterations
on seasonal ice cover and thermal structure of lakes are still rare.
Water clarity, which controls the amount of solar radiation penetrating into a lake, plays an
important role in heat budgets in lakes. Increased water clarity can result in warmer deep
waters (Stefan et al, 1996), while reduced light penetration can result in decreased mixing





depth and cooler deep waters (Hocking and Straskraba, 1999; Tanentzap et al., 2008).
Previous studies have also shown that increased water clarity was correlated with deeper
mixed-layer depth (Mazumder and Taylor 1994; Fee et al. 1996; Schindler et al. 1996) and
increased hypolimnetic heating rate (Yan 1983). Water clarity has important consequences for
photosynthesis and vertical distribution of biota. Accompanied by a warming climate,
increased evaporation and changes in precipitation patterns can alter the inputs of nutrients
and dissolved organic carbon (DOC) into lakes, resulting in changes in water clarity
(Schindler et al. 1990; Schindler et al. 1996).
Abrupt shifts, rather than linear changes, can occur in both climate and lake variables over
long time scales. In the past, the majority of referenced studies used some form of linear
regression analysis to look for changes, which assumes that lakes undergo monotonic changes
over time (Van Cleave et al., 2014). This assumption can mask the occurrence of step changes
(Liu et al., 2013; North et al., 2013) and can hide the underlying mechanisms that are
responsible for the changes (North et al., 2014). As noted in other studies (Mueller et al.,
2009; Van Cleave, 2014), long-term changes in lake thermodynamic variables can be
associated with a pronounced, nonlinear step change that indicates a rapid switch between
stable states or regimes (Scheffer et al., 2001; Rodionov, 2004; North et al., 2013). Identifying
this type abrupt change in climate drivers and corresponding abrupt or gradual change in lake
variables may shed light on the role of changing climate on the corresponding ecosystem.
The purpose of this study is to investigate how long-term changes in air temperature, wind
speed, and water clarity affect the ice cover and thermal structure of a dimictic Lake Mendota,
Wisconsin, USA during the past century using a long-term 104-year simulation model. We
hypothesize that changes in lake ice cover (ice-on and ice-off dates, ice cover duration, and
maximum ice thickness) and thermal structure (stratification onset, fall turnover date,
stratification duration, water temperatures, hypolimnetic heating) variables may be
characterized by periods of abrupt change rather than gradual trends, based on observations of
rapid change in the climate drivers of air temperature and wind speed. To address this, a one-
dimensional hydrodynamic model with ice cover is first validated using a long-term (104-
year) observational dataset. The model is then employed to simulate long-term (1911-2014)
ice cover and water temperature in the lake. With the knowledge of lake responses to these
past conditions, we aim to reveal how lakes might respond to future changes in air
temperature, wind speed, and water clarity.




# 2 Methods

## 2.1 Hydrodynamic model

An ice and snow model was added to the DYRESM-WQ model (Hamilton and Schladow 1997), a physically-based one-dimensional hydrodynamic model for simulating vertical mixing and advective transport in lakes and reservoirs. The resulting model, DYRESM-WQ-I, is validated and employed to simulate daily changes in the vertical distribution of water temperature and ice cover in Lake Mendota from 1911-2014. In this model, the lake is represented by a series of Lagrangian horizontal layers with uniform properties that may change in elevation and thickness in response to inflows/outflows and surface mass fluxes (evaporation and precipitation). Layer thickness is updated using an algorithm to give appropriate vertical density resolution at each time step. Mixing in the model is represented by merging the layers when the sum of available turbulent kinetic energy (TKE) produced by wind stirring, convective turnover, and shear stress exceeds the potential energy required to mix the adjacent layer below. Hypolimnetic mixing is modeled with an eddy diffusivity coefficient, which is a function of the dissipation of TKE and strength of stratification. More detailed descriptions of the simulation of water temperature and mixing are provided by Imberger and Patterson (1981).

The ice model is based upon a quasi-steady state assumption that the time scale for heat conduction through the ice is short relative to the time scale of meteorological forcing (Patterson and Hamblin, 1988; Rogers et al., 1995). This assumption is valid under a Stefan Number <0.1 (Hill and Kucera, 1983). The ice module is applied when the simulated surface water temperature first drops below 0 ºC; the initial ice thickness is set to a value of 5 cm (Patterson and Hamblin, 1988; Vavrus et al., 1996). Upward conductive heat flux between ice/snow cover and the atmosphere, $q_0$, is determined by numerically solving the quasi-steady state heat conduction equations (Rogers et al., 1995) and assigning appropriate boundary conditions to the water, ice, and atmospheric interfaces. At the ice (or snow) surface, a heat flux balance provides the condition for surface melting, and accretion or ablation of ice is determined through the heat flux at the ice-water interface. Imbalance between heat conduction through ice and the heat flux from the water to the ice gives the rate of change of ice thickness at the ice-water interface. Snow conductivity is estimated from its density using an empirical equation (Ashton, 1986), and snow compaction is based on an



exponential decay formula (McKay, 1968), with snow compaction parameters based on air
temperature and snowfall/rainfall (Rogers et al., 1995). Snow (white) ice is generated in
response to flooding, when the mass of snow that can be supported by the ice cover is
exceeded. The uppermost solid layer of ice or snow is adjusted in thickness at the end of each
1-hr model time step according to the balance of the heat budget. When ice thickness
decreases to less than 5 cm, open water conditions are restored.
Sediment heat flux, the main external source of heat after freezing, is important to water
temperatures beneath ice cover (Ellis et al., 1991). Sediment heat flux (Fang and Stefan,
1996) is included as a source/sink term for each Lagrangian layer to closely simulate under-
ice water temperatures. A simple diffusion relation (Rogers et al., 1995) is used to estimate
heat transfer from the sediments to the water column, $q_{sed}$:
$$q_{sed} = K_{sed} \frac{\mathrm{d}T}{\mathrm{d}z} \,. \tag{1}$$
where $K_{sed}$ is the sediment conductivity (=1.2 W m$^{-1}$ $^{\circ}$C$^{-1}$). d$T$/d$z$, the temperature gradient
across the sediment-water interface, is estimated as:
$$\frac{\mathrm{d}T}{\mathrm{d}z} \approx \frac{T_s - T_w}{z_{sed}} \,. \tag{2}$$
where $T_s$ is the sediment temperature, $T_w$ is the water temperature adjacent to the sediment
surface, which varies hourly and with depth, and $z_{sed}$ is the distance beneath the water-
sediment interface at which the sediment temperature becomes largely invariant. From data
collected at four locations on Lake Mendota (Birge et al., 1927), it was found that sediment
temperatures varied little at 5 m depth below the sediment-water interface, so $z_{sed}$ is set to be 5
m. In addition, data from Birge et al. (1927) is used to fit a curve to describe the seasonal
variation of $T_s$
$$T_s = 9.7 + 2.7 \sin\left[\frac{2\pi(D-151)}{TD}\right] \tag{3}$$
where D is the number of days from the start of the year and TD is the total number of days
for the year of interest (365 or 366). The vertical transfer of heat in the water column beneath
the ice is regulated by an assigned thermal diffusion coefficient. The formulation is the same
as that originally used in DYRESM-WQ (Hamilton and Schladow, 1997) to simulate heat



transfer throughout the open water period. This produced diffusivities that were within the
range of measurements by Ellis et al. (1991) of 1-3 times greater than molecular values.
Input for the model includes lake morphometry (lake volume and surface area as a function of
elevation), initial vertical profiles for water temperature and salinity, Secchi depth,
meteorological variables, and inflows/outflows. The model calculates the surface heat fluxes
using meteorological variables: total daily shortwave radiation, daily cloud cover, air vapor
pressure, daily average wind speed, air temperature, and precipitation. During the simulation,
all parameters/coefficients in the model are kept constant. The time step in the model for
calculating water temperature, water budget, and ice thickness was set to 1 hr. Snow ice
compaction, snowfall and rainfall components are updated at a daily time step, corresponding
to the frequency of meteorological data input. Cloud cover, air pressure, wind speed, and air
temperature are assumed to be constant throughout the day, and precipitation is assumed
uniformly distributed. Shortwave radiation distribution throughout the day is computed based
on the lake latitude and the day or year. The DYRESM-WQ-I model is calibrated using
measured lake variables including water level, temperature profiles, ice thickness, and ice on
and ice off dates. The overall simulation period was 104 years, starting with an isothermal
(measured) water column temperature of 3.1 °C on 7 April 1911 and ending on 31 October

18  2014.

## 2.2  Piecewise regression algorithm

Breakpoints in the air temperature trend over the study period were determined using a
piecewise linear regression (PLR) method (Toms and Lesperance, 2003; Tome and Miranda,
2004, Ying et al, 2015) that assumes continuity in the trendlines across the breakpoint. The
piecewise linear regression model finds the breakpoint, B, that minimizes the residual sum of
squares ($RSS_{PLR}$) of the model between the two phases (Ying et al, 2015)

$$RSS_{PLR} = \sum_{i=1}^{n} \left[ y_i - a - k_1 x_i - (k_2 - k_1) \max(x_i - B, 0) \right]^2 \qquad (4)$$

where $x_i$ and $y_i$ are the time and air temperature data corresponding to the $i$-th data point,
respectively, and $k_1$ and $k_2$ are the slopes of the linear fit to data before and after the
breakpoint $B$. The parameter $a$ is the intercept of the linear fit to data below the estimated
breakpoint $B$, and $n$ is the series length.





An $F$-test was used to compare the residual sum of squares of an ordinary linear regression
model with the piecewise linear regression model with a level of $\alpha = 0.05$ used to indicate
significance, and the breakpoint year was determined to pass with $p = 0.0016$.
**2.3   Sequential t-test analysis of abrupt changes**
Abrupt changes in mean annual wind speeds and lake ice cover and temperature variables
were detected using the sequential t-test STARS (Rodionov, 2004), which can automatically
detect multiple change points. A shift occurs when a statistically significant difference exists
between the mean value of the variable before and after a certain point based on the t-test. The
variables were tested using a threshold significance level $p = 0.05$, a Huber weight parameter,
h = 1 (Rodionov, 2006; North et al, 2013), and a cut-off length L = 15 years (North et al,

11  2013).

**3   Model input data**
**3.1   Bathymetry**
Lake Mendota (43°40'N, 89°24'W) has a surface area of 39.4 km$^2$ and maximum fetch of 9.8
km (Kitchell, 1992). The mean depth is 12.7 m, and maximum depth is 25.3 m. For detailed
bathymetry refer to Kamarainen et al. (2009).
**3.2   Meteorological variables**
Daily meteorological data required to run the DYRESM-WQ-I model include solar radiation,
air temperature, vapor pressure, wind speed, cloud cover, rainfall, and snowfall.  Daily air
temperatures are computed as the average of daily maximum and minimum temperatures.
Vapor pressure, wind speed, and cloud cover are also entered as daily average values while
solar radiation, rainfall, and snowfall are daily accumulation values.
Meteorological data for the Madison area have been continuously recorded since 1869,
however, the station and techniques have changed several times. Robertson (1989)
constructed a continuous, homogeneous daily meteorological dataset from 1884 to 1988 by
adjusting for changes in site location and observation time, and resultant changes in the
surface roughness (e.g. height of surrounding trees and buildings). These data were appended
with the most recent weather station of the National Climate Data Center (NCDC, NOAA)





located in Madison (MSN) Dane County Regional Airport (Truax Field), the same site as that
used in 1988. Since Robertson (1989) adjusted all historical data to that collected in 1988, no
adjustments are applied to the recent data except for wind. In 1996, a discontinuity in the
wind record was caused by change in observational techniques and sensor locations (McKee
et al. 2000). To address the non-climatic changes in wind speed, data from MSN are carefully
compared with those collected from the tower of the Atmospheric and Oceanic Science
Building       at       the       University       of       Wisconsin-Madison
(http://ginsea.aos.wisc.edu/labs/mendota/index.htm). Hourly data from both sites ($U_{MSN,hourly}$
and $U_{AOS,hourly}$) during 2003–2010 were used to form a 4×12 (four components of wind
direction × 12 months) matrix ($K_{4,12}$) of wind correction factors, yielding $U_{AOS,daily}=$
$K_{i,j}×U_{MSN,daily}$. A comparison of results indicated that the MSN weather station demonstrated a
higher magnitude in winds out of the east by 5% and lower magnitude in winds out of the
west and south by 30% and 10%, respectively. The adjusted wind data ($=K_{i,j}×U_{MSN,daily}$) are
employed and used in the model simulation. Overall the adjusted wind data show a decline in
mean wind velocities of 16% from 1988−93 to 1994−2014) compared to 7% at a nearby
weather station with no known observational changes (St. Charles, Illinois; 150 km southeast
of Lake Mendota).
**3.3   Light extinction**
Seasonal Secchi depths can be used to determine the light extinction coefficient for
DYRESM-WQ-I. Lathrop et al. (1996) compiled Secchi depth data for Lake Mendota
between 1900 and 1993 (1,701 daily Secchi depth readings from 70 calendar years), and
summarized the data for six seasonal periods: winter (ice-on to ice-out), spring turnover (ice-
out to 10 May), early stratification (11 May to 29 June), summer (30 June to 2 September),
destratification (3 September to 12 October), and fall turnover (13 October to ice-on). After
1993, Secchi depths were obtained from the North Temperate Lakes – Long Term Ecological
Research (NTL-LTER) (https://portal.lternet.edu/nis/home.jsp#). For years with no Secchi
data, the long-term mean seasonal Secchi depth was used to estimate light extinction. Light
extinction coefficients are calculated as a function of Secchi depth using the equation,
$k = 1.1/ z_s^{0.73}$ (Williams et al. 1980), where $k$ is the light extinction coefficient and $z_s$ is the
measured Secchi depth, in meters.



### 3.4 River inflow and outflow

Daily inflow measurements have been made on selected Lake Mendota tributaries since 1974, and daily outflow from the lake has been measured since 1975. These measurements were used to calculate total daily inflow and outflow. Total daily inflow and outflow from 1930 to 1974 were estimated from measurements made at the gauging station downstream of the lake using the drainage-area ratio method (Maidment 1993). Prior to 1930, total inflow and outflow were calculated using a water budget approach, i.e., the balance of inflow/outflow, precipitation, evaporation, and lake level changes. The inflow/outflow is the residual unknown term of the water balance where other terms include evaporation, rainfall, and water level. The calculation is performed at the interval of water level measurements and the residual term is then distributed evenly across the number of days between water level measurements. Water level of Lake Mendota has been recorded since 1916 at the Yahara River inlet in the north of the lake. Prior to 1916, the long-term mean lake level was assumed in water budget calculations. River water temperature has been recorded at the Yahara River inlet since 2002 (U.S. Geological Survey, http://waterdata.usgs.gov/nwis). To estimate daily river temperatures prior to 2002, river temperatures were estimated from air temperatures. A linear regression was used to correlate river temperatures and weekly average air temperature data during 2002–2009 for air temperatures above the freezing temperature ($r^2 = 0.86$), and a separate polynomial regression analysis was performed for air temperatures below the freezing temperature ($r^2 = 0.68$).

### 3.5 Lake ice cover and water temperature

Observed data from 1911-2014 (available at NTL-LTER website: https://portal.lternet.edu/nis/home.jsp#), showed that Lake Mendota, on average, freezes on 22 December (ice-on date), breaks up on 31 March (ice-off date), and has an ice duration of 98 days. Ice thickness and snow depth during the study period were compiled from various sources: 1911–1916 (unpublished data from E. Birge, University of Wisconsin); daily ice thickness during 1961–1962, 1962–1963, and for part of the winter 1963–1964 (Stewart 1965); 1975–1995 (unpublished data, D. Lathrop, Wisconsin Department of Natural Resources); after 1995, ice thickness and snow depth were sampled once or twice every winter by the NTL-LTER Program (https://portal.lternet.edu/nis/home.jsp#). In addition, during winters of 2008-2009 and 2009–2010, blue ice, snow ice, total ice (blue ice + snow ice), and snow cover depth were measured weekly at multiple locations on the lake (Hsieh,





2012). In total, there are 251 measurements of total ice, 21 for blue ice, 21 for snow ice, and
49 for snow depth during 1911 to 2014.
Long-term water temperature records were obtained from Robertson (1989) and the NTL-
LTER dataset (https://portal.lternet.edu/nis/home.jsp#). The frequency of water temperature
data varied widely from only one or two profiles per year to several profiles for a given day,
while there were no data collected in some years between 1931 and 1970. The vertical
resolution of the water profiles varied from 0.5 m to 2 m and sometimes 5 m when the water
column was weakly stratified.
**4   Results**
**4.1   Shifts in air temperature, wind speed, and water clarity**
Annual air temperature (Fig. 1a) had a relatively small increase from 1910 until 1980, but has
increased dramatically since 1981. Based on a piecewise linear regression algorithm, there
was a small warming trend of 0.081°C per decade during 1911−1980, followed by a dramatic
change (a warming trend of .334°C per decade) from 1981−2014. Figure 1b shows that mean
annual wind speed was 4.44 m s$^{-1}$ until 1994, when a significant shift occurred to 3.74 m s$^{-1}$
(15% reduction) based on the sequential t-test STARS method (Rodionov, 2004). Figure 1c
shows Secchi depth for Lake Mendota; however, no statistical trend was obtained due to the
incompleteness of the data.
Combining the statistically significant breakpoint in air temperature trend that occurred in
1981 and the shift in wind speed in 1994, the Madison climate may be broken into three
different periods. The first, from 1911−1980, was a relatively cool period and had an average
wind speed of 4.44 m s$^{-1}$. The second period (1981−1993) occurred after the breakpoint in the
air temperature trend and had a warmer air temperature and a wind speed of 4.44 m s$^{-1}$. The
third period (1994−2014) occurred after the shift in wind speed from 4.44 m s$^{-1}$ to 3.74 m s$^{-1}$
and had even warmer air temperatures.
**4.2   Model validation and long-term simulations**
Here we provide results of model validity comparisons followed by results of long-term
changes in both observed (where available) and simulated conditions for: snow and ice





thickness, ice-on and ice-off dates, lake water temperature, summer stratification, and
thermocline depth.

### 4.2.1 Ice and snow thickness

Figure 2 shows the comparison of the long-term (from 1911 to 2014) simulation and
measured data for blue ice, white ice, and total ice thickness and snow depth. Mean absolute
differences between simulated and measured thickness of total ice, blue ice, snow ice, and
snow cover are 7.8 cm (n=251; RMSE=7.9 cm), 5.5 cm (n=21; RMSE=3.3 cm), 1.9 cm
(n=21; RMSE= 2.3 cm), 4.1 cm (n=49; RMSE= 3.7 cm), respectively. Some discrepancy
between the model results and measurements may be due to the 1-D structure of the model.
The DYRESM-WQ-I model provides only the lakewide average ice and snow depths, hence
no horizontal variation. For a medium-to-large lake, such as Lake Mendota, spatial variations
in ice and snow depths should be expected (Bengtsson 1986). In the winters of 2008–2009
and 2009–2010, ice measurements were made at multiple locations on Lake Mendota (Hsieh
2012). Data in 2008–2009 showed variations in ice thickness. Total ice thickness in the
middle of the lake was approximately 12 cm greater than that at a littoral location (water
depth=2 m) in late winter. In 2009–2010, ice cover was fairly spatially uniform through the
entire sampling period. Consequently, discrepancy between model results and measurements
is expected in those years when ice thickness varied spatially. The accuracy of predictions
from DYRESM-WQ-I was compared with other models: MINLAKE (Fang and Stefan 1996)
produced standard errors of 11 cm for total ice thickness and 6 cm for snow cover for Thrush
Lake, Minnesota, and 12 cm for total ice and 7 cm for snow cover for the north basin of Little
Rock Lake, Wisconsin. Other models including LIMNOS (Vavrus et al. 1996) on Lake
Mendota, Wisconsin; MLI (Rogers et al. 1995) on Harmon Lake, British Columbia; and
CLIMo (Duguay et al. 2003) on lakes in Barrow, Alaska; Poker Flat, Alaska; and Churchill,
Manitoba produced similar errors to Lake Mendota between modeled and observed ice
thickness and snow cover. In general DYRESM-WQ-I predicts the ice and snow depths fairly
accurately. Figure 3a shows the evolution of simulated ice and snow thickness with multiple
ice and snow thickness measurements taken during the winter of 2009-2010. Once ice
develops, water beneath the ice continues to freeze, adding to the ice thickness, as heat is
conducted from water to the air. In this year, lake ice grew fast initially and then slowed
substantially. Eventually, the ice reaches its maximum thickness and stops growing (Ashton
1986). Once air temperatures begin to warm up, ice quickly thins until ice out. Overall, the



model is able to capture the growing ice cover as well as the temporally variable snow cover
found on the lake.
Figure 3b shows the long-term (1911–2014) simulated annual maximum ice thickness during
each winter. The annual maximum thickness varied widely from 20.1 cm (1997–1998) to 72.0
cm (1911–1912) during the 1911–2014 period. The timing when ice reached its maximum
thickness also varied considerably, from 17 January (1973) to 7 April (1923). In 73 out of 103
winters, ice reached its maximum thickness in March. Linear regression analysis on simulated
maximum ice thicknesses indicates that it has decreased at a rate of 12.7 cm per century
during 1911–2014. A t-test of the mean values shows a statistically significant ($p<0.05$)
difference in the mean annual maximum ice thickness between period 1 (1911−1980) and
period 3 (1994−2014). Period 2 was not statistically different from either of the other two
periods.

### 4.2.2  Ice-on and ice-off dates

Figure 4 shows measured and simulated ice-on date, ice-off date, and ice duration on the lake.
The measured ice-on date is defined as the first day when the lake becomes fully ice-covered,
and the ice-off date is the last day of the ice breakup before the open water season. Simulated
results and observations are in good agreement, with a mean absolute difference of 2.3 days
(RMSE =2.4 days) for ice-on date and 5.7 days (RMSE = 4.8 days) for ice-off date. It is noted
that both the mean error and RMSE are much smaller than the interannual variability in the
observed ice-on dates (standard deviation SD = 11.2 days) and ice-off dates (SD = 10.6 days).
Interannual variations in ice cover on Lake Mendota were large in the past century. Observed
ice-on dates ranged from 3 December (1929) to 30 January (1931): a range of 58 days. Ice-off
dates ranged from 27 February (1998) to 20 April (1923): a range of 51 days. The model
successfully captures the interannual variations of ice-on and ice-off dates. For example, it
reproduces the unusual late ice-on date in 1930-1931, and several noticeable early ice
breakups associated with intense El Niño -Southern Oscillation (ENSO) events, i.e., 1965,
1972, 1982, and 1997 (Anderson et al. 1996; Magnuson et al. 2000; Robertson et al. 2002).
Figure 4b shows the ice duration, defined as the period between ice-on and ice-off dates. The
mean absolute difference in ice duration between the model results and observations is 6.6
days (RMSE = 6.1 days), compared to the observed SD of 17.9 days.  Overall, we consider
the model performs well in simulating ice on and off dates and ice cover duration.





Long-term trends in ice formation are examined by applying linear regression to the model
results and observed data. Figure 4 clearly shows progressively later freezing, earlier breakup,
and shorter duration in Lake Mendota from 1911 to 2014. Based on model results, ice-on
dates became later by 9.0 days, ice-off dates became earlier by 12.3 days, and ice duration
shortened by 21.3 days. Model results are in good agreement with those obtained from the
observed data (7.4 days later ice-on dates, 9.3 days earlier for ice-off dates, and 18.0 days
shorter duration). All linear trends from the observed and simulated data are statistically
significant ($p < 0.05$). Mean values of ice-on, ice-off, and ice duration for the three selected
periods, i.e., 1911−1980, 1981−1993, and 1994−2010, (see Table 1 and Fig. 4) show a
statistically significant ($p < 0.05$, based upon t-test) difference between period 1 (1911−1980)
and period 3 (1994−2014) for all ice cover variables and a statistically significant ($p < 0.05$)
difference between period 1 and period 2 (1981−1993) for ice cover duration.

### 4.2.3 Water temperature

The performance of the model in simulating water temperatures in the lake is presented in
Figures 5 and 6. For temperatures at the near-surface, we use the simulated volume-weighted
mean water temperatures between depths of 0 and 10m. The simulated epilimnetic
temperatures compare well with those estimated from the measured data; the annual mean
absolute error is 0.69 ºC (n=3,239; RMSE=0.30 ºC), demonstrating the ability of the model to
simulate the heat budget in the upper mixed layer of the lake. Near-surface temperatures range
from about 0 ºC in winter to 26.1 ºC in summer, and strongly respond to the net surface heat
flux, as illustrated by the 2–3 ºC variations that occur in both simulated and measured data,
e.g. in the summer of 1921, 1960, 1994, and 2009 (Fig. 5). Overall the model captures
interannual variations in epilimnetic temperatures even during extreme years; for example, the
relatively high maximum epilimnetic temperatures (> 25 ºC) in the summers of 1949, 1955,
1963, 1987, 1988, and 1999, and relatively cold annual maximum epilimnetic temperatures (~
22 ºC) in the summers of 1915, 1924, and 1942.
Figure 6 shows the comparison between the simulated volume-weighted near-bottom (within
the hypolimnion) temperatures over the depths between 20−25 m and those estimated from
the measured data. The mean absolute error is 1.04 ºC (n=3,239; RMSE=0.53 ºC). Near-
bottom temperatures range from 0.2ºC in winter to 19.1ºC in autumn, which is approximately
10 ºC less than the variation in near-surface temperatures. Near-bottom temperatures are
lowest when the lake first freezes, and then slowly increase because of heat released from the





bottom sediment although we cannot discount that cold inflows could also penetrate into the
lake and affect temperature at the selected depths (20-25 m).After ice break-up and just prior
to seasonal stratification, temperature in near-bottom waters is usually identical to surface
waters and the whole water column undergoes a period of sustained increase in temperature.
Hypolimnetic temperatures then stay relatively constant during the stratified period with
limited heat exchange associated with strong temperature gradients across the thermocline.
To further examine the model's capability of simulating the extreme differences in
temperature, we compare measured and modeled temperatures under years with cold (1924)
and warm (1963) epilimnion temperatures and cold (1972) and warm (1926) hypolimnion
temperatures. Figure 7 shows that the model accurately captures the extreme conditions in
those four years. For plotting, measurement values are assumed to remain constant for the
duration of time between measurements. Measurements are generally taken at 1 m intervals
over the depth of the water column, but in instances with irregular measurement intervals,
temperatures are interpolated at intervals of 1 m for the depth of the lake. Slight variation in
temperatures at each depth may be attributed to differences between the averaged seasonal
Secchi depth values used to drive the model and the observed Secchi depth values. To
demonstrate the model's ability to simulate changes in ice conditions and water temperature
under the ice, measured and simulated conditions for these extreme years are presented in Fig.
8. In this figure, the time frame and temperature scales are adjusted to highlight the subtle
changes in winter and ice conditions and modeled ice and snow thickness are added to the top
of the plots. While some discrepancies in temperature are recognized in the model
comparison, the model generally captures the inverse stratification under the ice cover.
Updated measurements of sediment temperature during winter may provide a better estimate
of the sediment heat flux during ice-cover conditions, and improve under-ice temperature
predictions. Overall, the model can reliably simulate water temperature in open water and ice
seasons for extreme cold and warm years.
Mid-summer (averaged between 16 July and 15 August when the thermocline was well-
established) epilimnetic (0−10m) and hypolimnetic (20−25m) simulated temperatures from
1911 to 2014 are shown in Fig. 9a. Epilimnetic temperatures range from 19.7 ℃ to 24.8 ℃.
There is no statistically significant temporal trend in the mid-summer epilimnetic
temperatures, which may be partly explained by the lack of a warming trend for the mid-
summer air temperature in the Madison area based on the measured meteorological data





(Kucharik et al. 2010). In contrast, hypolimnetic temperatures range from 7.8 ºC to 16.2 ºC.
The interannual variation in hypolimnetic temperatures (SD = 1.74ºC) is greater than that in
epilimnetic temperatures (SD = 1.0ºC). Simulated hypolimnetic temperatures show a
decreasing trend of -1.4 ºC per century ($p<0.05$). Figure 9b shows that the decreased
hypolimnetic temperatures resulted in the epilimnion-hypolimnion temperature difference
increasing by 2.0 ºC per century ($p<0.05$). Mid-summer hypolimnetic temperature and
epilimnion (0-10 m)-hypolimnion (20-25 m) temperature difference both exhibit significant
($p<0.05$) mean differences between period 1 (1911−1980) and period 3 (1994−2014), as
shown in Table 1.
Figure 10 shows the simulated long-term (104-year) mean under-ice water temperature
(volume-weighted average of all depths) ranging from 0.78 ºC to 3.07 ºC (mean = 1.74 ºC;
SD = 0.54 ºC). Based on model results, under-ice temperatures reveal a significant increasing
trend ($p<0.05$) of 0.48 °C per century during 1911–2014, suggesting that air temperature and
wind speed may affect under-ice water temperature. For the selected regimes, the under-ice
(bold lines) and freeze-over (dashed lines; volume-weighted average of all depths) water
temperatures exhibit interesting features. The difference between the water temperature over
the entire ice-covered period and the freeze-over water temperature has decreased through
time, indicating less heat gain during the winter, which may relate to a shorter time to transfer
heat and a smaller gradient in temperatures between the lake and the bottom sediments. The
under-water ice temperatures and freeze-over water temperatures both show significant
($p<0.05$) differences between regime 1 (1911-1981) and regime 3 (1994-2014), as shown in
Table 1.

### 4.2.4 Summer stratification period and thermocline depth

We characterize summer stratification by the dates of the onset of stratification and fall
turnover, and the total duration of stratification. Accurately describing these conditions
requires high frequency observations, which is challenging for long-term datasets. As a result,
we used modeled data to determine these variables. The dates of stratification onset/fall
turnover are defined as the day when the surface-to-bottom temperature difference is
greater/less than 2 ºC (Robertson and Ragotzkie 1990). Figure 11 shows the dates of onset of
stratification breakdown (fall turnover), and the total duration of stratification. The onset dates
varied between 17 April (1977) and 30 June (1981), and the date of fall turnover varied from
02 August (1965) to 19 October (2013). Over 1911–2014, the onset of thermal stratification
has become earlier by 12.3 days ($p<0.05$), and fall turnover has become later by 14.6 days
($p<0.05$), resulting in the stratification period increasing by 26.8 days ($p<0.05$). Stratification
onset date shows no significant difference among the three regimes (see Table 1); however,
fall turnover date is significantly ($p<0.05$) different between periods 2 and 3 and between
period 1 and 3, and the duration of stratification is significantly ($p<0.05$) different between
periods 1 and 3, as shown in Table 1.
The thermocline depth, defined as the depth of maximum temperature gradient, is determined
from the model results (not shown in figure for brevity). During 1911−2014, the mid-summer
thermocline depth in Lake Mendota varied from 8.3 m to 12.4 m (mean = 10.7 m; SD = 0.84
m) with no significant long-term change.
## 5   Discussion
### 5.1   Significance of lake drivers
Changes in long-term simulated ice cover and thermal variables in Lake Mendota from
1911−2014 appear to occur over short time frames within this period, synchronous with rapid
changes in measured drivers (i.e., air temperature, wind speed, and water clarity). The
variables examined included ice on/off dates, maximum ice thickness, freeze-over water
temperature, mid-summer epilimnetic and hypolimnetic temperatures, summer hypolimnetic
heating, and dates of stratification onset and fall turnover. To identify the the relationship
between the drivers and each of these variables, we employed Pearson correlation analysis on
the detrended driver and simulation data. While using this method does not allow us to
directly determine causality, we may identify possible related drivers to changes in lake
variables and the relative importance of relationship of the three drivers (i.e. air temperature,
wind speed, and water clarity) to the lake variables. To calculate correlation coefficients, each
lake driver was averaged over a fixed period (e.g. April-May or November-December) and
then paired with each of the simulated lake variables from the same period. The averaging
period for air temperature and wind speed was chosen based on the fixed period that yielded
the best correlation. We chose to employ fixed periods because thorough testing indicated that
using a dynamic period introduced seasonal meteorological variations into the analysis. For
instance, the correlation coefficient between onset date and the mean air temperature averaged
over the prior month is expected to be high simply due to the seasonal variations over the





course of the year, i.e., the later in the spring, the warmer air temperature. Seasonal Secchi
depths were compiled for six seasonal periods: winter, spring turnover, early stratification,
summer, destratification, and fall turnover (Lathrop et al. 1996), and the period closest to the
averaging period used for air temperature and wind speed was used for correlation analysis.
The averaging periods used for calculating correlation coefficients of each pair of variables
are listed in Table 2. We discuss below the correlations for each pair of variables, shown in
Figure 12.

### 8    5.1.1 Ice cover variables

Ice-on and ice-off dates have been shown to be sensitive to climate conditions (Robertson et
al. 1992; Livingstone 1997; Magnuson et al. 1997). For both air temperature and wind speed,
the averages from 1 November to 31 December (ND) give the highest correlation with ice-on
date; in agreement with the results of Assel and Robertson (1995) and similar to Gao and
Stefan (1999). Averaged air temperature and wind speed over the period from 1 February to
31 March (FM) provide the highest correlation with ice-off date, consistent with the study by
Gao and Stefan (1999), who found that ice-off dates for ten Minnesota lakes correlated
significantly with February to April air temperatures. The strong positive correlation ($r =$
0.82) between ice-on date and N-D air temperature indicates a relationship between warmer
N-D air temperatures and later freezing dates. Strong negative correlation between ice-off
date and F-M air temperature ($r = -0.70$) indicates a realationship between earlier ice breakup
and warmer F-M air temperatures. In contrast to air temperature, ice cover is only weakly
related to the seasonal average wind speed. Nevertheless the correlation between ice-on date
and wind speed ($r = -0.26$) is significant, indicating a relationship between decreasing wind
speed and earlier ice-on. Since large wind events prevent ice formation by breaking up skim
ice, decreased wind speeds allow ice cover to form slightly earlier in the year. In the Madison
area, the decreasing wind speeds may act to slightly mitigate the effects of increasing air
temperature in later ice freezing. Correlation between ice dates and Secchi depths is low ($r =$
0.24 for ice on and $r = -0.22$ for ice off), indicating that water clarity has only a minor
relationship with ice dates, in agreement with  the simulation results of Fang and Stefan
(1997). For both ice-on and ice-off dates, correlations with wind speed and Secchi depth are
much less significant than with air temperature, indicating that air temperature has a more
significant relationship with ice-on and ice-off dates and may be a more significant indicator
of ice cover dates and ultimately ice cover duration.





Annual maximum ice thickness is negatively correlated with the 1 January – 31 March (J-F-
M) averaged air temperature ($r = -0.77$) and weakly correlated with J-F-M wind speed ($r =$
0.21), indicating a better relationship between annual maximum ice thickness and air
temperature than for wind speed. The relationship of air temperatures on ice phenology is
consistent with previous findings for lakes in Northern Wisconsin, Canada, and Maine (Gao
and Stefan 2004).
The strong correlations between air temperature and ice cover variables agree with several
earlier studies (Anderson et al. 1996; Goa and Stefan 2004; Vavrus et al. 1996; Livingstone
1997; Williams et al. 2004). Snowfall has also been shown to be related to ice-off date
(Jensen et al. 2007) and ice thickness (Vavrus et al. 1996; Duguay et al. 2003) because
snowfall changes surface albedo and isolates ice cover from the atmosphere. We found
simulated number of snow days (number of days when the lake has snow cover) is strongly
correlated with ice-off date ($r = 0.70$) and maximum ice thickness ($r = 0.75$), indicating
snowfall is also significantly related to ice cover.
## 5.1.2  Water temperature and stratification
Figure 10 shows that the simulated water temperatures in Lake Mendota at the time of
freezing have been increasing since 1911. The correlation between freeze-over water
temperature and lake drivers during 1 November and 31 December (N-D) is significant but
with low $r$ values for wind speed ($r = -0.37$) and air temperature ($r = -0.24$) and insignificant
for Secchi depth ($r = 0.18$). This result indicates that wind mixing is related to cooling of the
water column before Lake Mendota freezes.
The simulated onset dates of stratification are negatively correlated with April-May (A-M) air
temperature ($r = -0.55$) and positively correlated with A-M wind speed ($r = 0.35$). In Lake
Mendota, the combination of increasing air temperatures and decreasing wind speeds are both
related to earlier onset of stratification. Austin and Colman (2007) suggested that the
declining ice cover combined with higher air temperatures cause the earlier onset of
stratification in Lake Superior at a rate of 0.5 day yr$^{-1}$. However, no correlation between ice-
off date and the onset of stratification was found for Lake Mendota in this study.
Mid-summer (16 July – 15 August) epilimnetic temperatures are most strongly correlated
with the air temperature and wind speed averaged over the corresponding period. Epilimnetic
temperature is highly correlated with air temperature ($r = 0.77$) and not significantly



correlated with wind speed averaged over the corresponding period ($r = –0.11$) and Secchi
depth ($r = 0.09$). Mid-summer hypolimnetic temperatures depend on the timing of
stratification (how long the near-bottom water is heated before the onset of stratification) and
the amount of heat mixed into hypolimnion before and during the stratification period.
Hypolimnetic temperatures are weakly correlated with air temperatures ($r = -0.23$) and not
significantly correlated with wind speed ($r = 0.11$) and Secchi depth ($r = –0.02$). Summer
hypolimnetic heating (temperature change between 1 July and 31 August, J-A) is most
strongly correlated with J-A wind speed ($r = 0.49$), moderately correlated with Secchi depth
averaged over 30 June – 2 September ($r = 0.35$), and not significantly correlated with J-A air
temperature ($r = –0.02$). This indicates a relationship between stronger winds and clearer
water and deeper heat mixing.
Fall turnover occurs when the lake water cools, driven largely by colder air temperature and
wind-induced mixing. The date of fall turnover is significantly negatively correlated with
September wind speed ($r = -0.43$) so that higher wind speed is related to earlier fall turnover.
Air temperature ($r = 0.11$) and Secchi depth ($r = –0.15$) are not significantly correlated with
the timing of fall mixing. Interestingly, the mid-summer epilimnion-hypolimnion temperature
difference has a higher correlation with turnover date ($r = 0.62$) than any of the three lake
drivers. This finding suggests that the influences of spring conditions may be transmitted to
the following summer and fall seasons. In other words, the cooler hypolimnion and the greater
epilimnion-hypolimnion temperature difference after a warmer and less windy spring may be
related to later fall turnover and a longer stratification period.

## 5.2   Abrubt changes in lake variables

To investigate the effects of abrupt changes in air temperature and wind speed on lake ice
cover and water temperatures, we used the hydrodynamic model DYRESM-WQ-I to describe
changes in several lake variables during 1911−2014. Simulation results were used to examine
differences in mean values of these lake variables between specific periods (see Sect. 4.1).
For each period of the selected periods, mean lake variables were calculated and the
differences between periods were analysed with t-tests to determine if they were significantly
different. Table 1 lists the mean values and differences for the nine lake variables during the
three selected periods. Comparison of period 1 (1911-1980) to period 2 (1981-1993) of lake





variables shows a shift to warmer air temperature; period 2 to period 3 (1994-2014) represents
an abrupt change to lower wind speed; and period 1 to period 3 represents a shift in to warmer
air temperature combined with an abrupt change to lower wind speeds.

### 5.2.1 Ice cover

Three simulated ice cover variables (maximum ice thickness, ice-on date, and ice-off date)
show no significant difference in means between period 1 and 2. In other words, the abrupt
change in air temperature trend does not result in a different ice regime even though the ice
cover variables are all highly correlated with air temperature ($r >0.70$). This may be because
the change in air temperatures was not of sufficient magnitude to cause a particularly large
change in ice cover or it may signify that other drivers are contributing to changes in ice cover
variables. Additionally, no significant difference is observed between period 2 and period 3
for the ice cover variables since the wind speed and ice cover variables are only weakly
correlated. The ice variables do show a statistically significant difference in mean values
between periods 1 and 3, indicating that a significant shift in these variables occurs only after
a sufficiently large increase in air temperature and an abrupt shift in wind speed within the
time between periods 1 and 3. In other words, air temperature need to increase sufficiently to
observe a statistically significant difference in ice cover. Ice cover duration, however, shows a
significant difference in the mean between all regimes (1−2, 2−3, and 1−3), indicating that
distinct differences in ice cover duration can be affected by both trends in air temperature
(i.e., there was a large enough change in air temperature between each period) and an abrupt
shift in the wind speed. The combined effects of slightly later ice-on dates and earlier ice-off
dates during each of the three periods resulted in statistically significant difference in mean
ice cover duration values between each of the three periods.
Analysis of simulated maximum ice thickness, ice-on, ice-off, and ice cover duration using
the method of Rodionov (2004) shows that the most statistically significant timing of the shift
in these ice cover variables occurs in the winter of 1997-1998, but a major shift in the air
temperature or wind speed data was not observed at that time. The unusual winter of 1997-
1998 strongly drove the statistically significant difference in mean values between periods 1
and 3 rather than the abrupt shift in wind speed in 1994. Interestingly, similar results have
been reported in Lake Superior, where statistically significant step changes were found in
winter ice duration and maximum wintertime ice extent; these step changes account for most
of the long-term trends in ice cover for the lake (Van Cleave et al, 2014). This timing of this



may be attributed to a combination of the longer term changes in meteorological conditions
and the short-term annual change occuring in the warm El Niño winter of 1997-1998 (Van
Cleave et al. 2014). Mueller et al. (2009) found that a similar climate shift between 1997 and
1998 initiated a change in lake ice phenology from infrequent to frequent summer loss in
several high-Arctic lakes.

### 5.2.2 Water temperature and stratification

Means of five simulated lake variables (under-ice water temperature, freeze-over water
temperature, epilimnion-hypolimnion temperature difference (indicative of strength of
stratification), and duration of stratification) over the three periods have significant ($p<0.05$)
differences only between period 1 and period 3. This change likely occurs because of the
combined effects of large changes in air temperature and a change in wind speed. Both air
temperature and wind speed are significantly correlated with these five lake variables. Each
driver alone may not be strong enough to cause a major shift in the lake variables, but their
combined effects may reinforce the drivers of abrupt change in ice and thermal phenology.
Further work is required to examine how the major drivers may either reinforce or dampen
lake ice and temperature responses, particularly in relation to directional shifts predicted
under climate change.
Fall turnover date, highly correlated with wind speed, exhibits a significant ($p <0.05$) shift in
the mean value in 1994, corresponding with the abrupt shift in the wind speed. Interestingly,
hypolimnetic water temperatures, which are not significantly correlated with wind speed, but
are correlated with air temperatures, also show a significant ($p<0.05$) shift in the mean value
in 1994. Hypolimnetic heating, significantly correlated with wind speed ($r = 0.49$), does not
exhibit a significant breakpoint, nor are any of the mean differences among the three periods
significant. Given the high correlation between wind speed and hypolimnetic heating, it is
hypothesized that there should be a shift in hypolimnetic heating caused by the abrupt shift in
wind speed in 1994. The lack of statistically significant step change may be explained by the
simultaneous high correlation between Secchi depth and hypolimnetic heating ($r = 0.35$),
indicating that water clarity may act to inhibit heating regardless of changes in wind speed, or
it may be acting to filter or mitigate the effects of the wind speed shift. Finally, mean onset
date of stratification and mid-summer epilimnetic temperature exhibit no difference among
the three periods. This may be due to two processes: (i) the climate signal is being filtered out





the lake; or (ii) the external perturbation of the system is not yet strong enough to trigger a
major shift in the system's internal dynamics.
**5.3   Ecological significance of long-term changes in lake variables**
Changes in lake ice cover and thermal structure are of great ecological significance. Summer
stratification inhibits the vertical transport of oxygen and nutrients. The vertical temperature
gradient in Lake Mendota increased (Fig. 9b), which should decrease the depth of vertical
mixing and oxygen penetration and decrease the transport of nutrients from the hypolimnion
to the epilimnion (Fee et al. 1994). Thus, increases in air temperature may increase the
epilimnetic-hypolimnetic temperature gradient and decrease the extent of summer algal
blooms in the Lake Mendota where internal loading of phosphorus has been shown to be
significant to its summer phosphorus budget (Soranno et al. 1997; Lathrop et al. 1998).
Additionally, phosphorus entrainment predominately occurs during periods of high wind
(Stauffer and Armstrong 1986; Kamarainen et al. 2009). Algal blooms have been associated
with increases in epilimnetic phosphorus due to these high-wind episodes (Stauffer and
Lee1973; Soranno et al. 1997; Robarts et al. 1998). Changes in the frequency of high wind
episodic events may further impact the growth of algal blooms within Lake Mendota (Kara et
al. 2012). Elevated water temperatures can affect plankton community composition and
abundance (Elliot et al. 2005; Findlay et al. 2001; Francis et al. 2014; Rice, 2015) and fish
populations (Magnuson et al. 1990; Carpenter et al. 1992; Destasio et al. 1996; Gunn 2002).
High water temperatures have been also shown to enhance the dominance of cyanobacteria
(Huber et al. 2008; Johnk et al. 2008). Such changes in algal blooms, zooplankton
populations, and fish populations may have drastic effects on the overall food-web within
Lake Mendota (Carpenter et al. 2007). Variations in ice-cover conditions have impacts on
ecosystems by dramatically changing habitat factors such as light and temperature (Adrian et
al. 1999; Quayle et al. 2002). Such changes impact the distribution, behavior, community
composition, reproduction, and evolutionary adaptations of organisms (Sala et al. 2000).
Previous studies have shown that climate change will likely affect the biodiversity of
freshwater ecosystems (Wrona et al, 2006; Heino et al. 2009; Mantyka-Pringle et al. 2014).
Climate change can bring asynchronies in biotic communities that may result in a
phenological decoupling of trophic relationships (Stenseth and Mysterud, 2002).



## 6   Conclusions

The 1-D hydrodynamic-ice model, DYRESM-WQ-I, is developed and validated, then used to simulate the ice cover and thermal structure of Lake Mendota, WI from 1911−2014. The model reliably reproduces the interannual variations and long-term (104-year) changes in ice cover and thermal structure. To our knowledge, this study presents the first attempt to continuously model both ice cover and thermal structure of a lake over a period as long as a century. Simulated ice cover over this period has changed dramatically: freezing later (9.0 days per century), breaking up earlier (12.3 days per century), resulting in shorter ice duration (21.3 days per century), and thinner maximum ice thickness (12.7 cm per century). These results agree well with the observed data and previous studies. For lake thermal structure, results of the continuous model simulation indicate an earlier onset of thermal stratification (12.3 days per century), later fall turnover (14.6 days per century), and a longer stratification period (26.8 days per century) during this period. In addition, simulated mid-summer hypolimnetic temperature decreased (−1.4°C per century), causing mid-summer epilimnetic-hypolimnetic temperature differences (strength of stratification) to increase (2.0°C per century). Interestingly, there is no significant trend in mid-summer epilimnetic temperatures.

Correlations comparing simulated lake conditions with seasonally averaged meteorological conditions indicate that, among the three drivers, air temperature has the largest relationship with the ice cover variables (ice-on and ice-off dates, and maximum ice thickness) and with three stratification variables (date of the onset of stratification, epilimnetic temperatures, and hypolimnetic temperatures). Freeze-over water temperature, summer hypolimnetic heating, and data of fall turnover all have the largest correlation with wind speed. Both air temperature and wind speeds are highly correlated with the onset of stratification. Secchi depth does not appear to have a strong relationship with interannual variablity of these variables for Lake Mendota, but in combination with wind speed may be related to hypolimnetic heating.

Changes in meteorological factors over the past 104 years were examined to determine if there have been abrupt shifts, rather than linear changes. Based on a change in the trend of air temperature increase occurring in 1981 and a major shift in wind speed in 1994, the Madison climate is divided into three distinct periods: 1911−1980, with relatively low air temperatures and mean wind speeds of 4.44 m s⁻¹; 1981−1993, with higher air temperatures and mean wind speeds of 4.44 m s⁻¹; and 1994-2014 with still higher air temperatures and mean wind speed of 3.74 m s⁻¹. Ice cover duration exhibited a significant difference in the mean among all three





periods, while ice-on, ice-off, and maximum ice thickness only show a significant difference
between period one and three, indicating that only with a large change in air temperature and
an abrupt shift in wind speeds are change in the ice cover variables statistically different. Mid-
summer hypolimnetic temperature and fall turnover date both reveal significant ($p <0.05$)
differences in the mean value in 1994, corresponding with the abrupt shift toward lower wind
speeds. Some lake variables (under-ice water temperature, freeze-over water temperature,
epilimnion-hypolimnion temperature difference, and stratification duration) may not be driven
by either the change in air temperature trend or the abrupt shift in wind speed alone, but a
shift in the mean of the lake variables does occur in 1994 when both the air temperatures are
warmest and the wind speed experienced an abrupt shift.  The exact timing of shifts may be
difficult to define because of extreme changes in weather in specific years and it may mask
the longer term changes in meteorological conditions (i.e. abrupt shifts).
Linear trends from the extensive data collection may be misleading as they masks these
sudden shifts in the air temperature and wind speed drivers and lake ice cover and thermal
structure variables. Examining the mean differences in lake variables in response to shifts in
air temperature and abrupt shifts in wind speeds provide a better understanding of how the
Madison Area Wisconsin climate has changed and what impact those changes have on shifts
in the ice cover and water temperature of dimictic Lake Mendota. It is shown that air
temperature and wind speed changes have occurred in stages and lake variables have
responded in a non-linear way to these changes.
**Acknowledgements**
Financial support for this project was provided in part by the U.S. National Science
Foundation Long-Term Ecological Research Program, University of Wisconsin (UW) Water
Resources Institutes USGS 104(B) Research Project, and UW Office of Sustainability SIRE
Award Program. Funding support for the first author by the College of Engineering Grainger
Wisconsin Distinguished Graduate Fellowship is acknowledged. We also thank Yi-Fang
Hsieh further developing an ice module in the DYRESM-WQ model in work that was
initiated by Brett Wallace. Yi-Fang Hsieh also collected ice data for model calibration and
validation. Finally, the authors would like to thank John Magnuson and Tim Kratz at the
Center for Limnology for their insightful suggestions and valuable comments regarding
climate change on lake ice.





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



1    Table 1. Mean values of climate drivers and lake variables of three hypothesized periods during 1911−2014. Asterisks (*) mark significant

2    differences between two period ($p < 0.05$).

| Driver/Variable | Unit | Period | | | Difference in Mean | | |
|---|---|---|---|---|---|---|---|
| | | (1)1911-1980 | (2)1981-1993 | (3)1994-2014 | (1) and (2) | (2) and (3) | (1) and (3) |
| Lake driver | | | | | | | |
| Air temperature (slope) | ºC decade$^{-1}$ | 0.081 | 0.334 | 0.334 | | | |
| Wind speed | m s$^{-1}$ | 4.44 | 4.44 | 3.74 | | | |
| Lake variables | | | | | | | |
| Maximum ice thickness | cm | 49.8 | 44.9 | 40.7 | −4.9 | −4.2 | -9.1* |
| Ice-on date (model) | Date | 21-Dec | 23-Dec | 29-Dec | 2 days | 6 days | 8 days* |
|          (observation) | Date | 21-Dec | 24-Dec | 29-Dec | 3 days | 5 days | 8 days* |
| Ice-off date (model) | Date | 9-Apr | 2-Apr | 30-Mar | −7 days | −3 days | -10 days* |
|          (observation) | Date | 3-Apr | 27-Mar | 26-Mar | −7 days | −1 days | -8 days* |
| Ice duration (model) | Days | 108.7 | 99.5 | 91.1 | −9.2* | −8.4* | -17.6* |
|          (observation) | Days | 103.2 | 92.9 | 85.6 | −10.3* | −7.3* | -17.6* |
| Under-ice water temperature | ºC | 1.74 | 1.81 | 2.08 | 0.07 | 0.27 | 0.34* |
| Freeze-over water temperature | ºC | 1.03 | 1.14 | 1.66 | 0.11 | 0.52 | 0.63* |
| Stratification onset date | | 24-May | 17-May | 18-May | −7 days | −1 days | -6 days |
| Mid-summer epilimnetic temperature | ºC | 23.0 | 23.2 | 23.4 | 0.2 | 0.2 | 0.4 |
| Mid-summer hypolimnetic temperature | ºC | 12.0 | 11.8 | 10.9 | −0.2 | −0.9* | -1.1* |
| Epilimnion-hypolimnion temp. difference | ºC | 10.9 | 11.4 | 12.5 | 0.5 | 1.1 | 1.6* |
| Hypolimnetic heating (1 July−31 August) | ºC | 0.699 | 0.688 | 0.583 | -0.011 | −0.105 | -0.116 |
| Turnover date | Date | 20-Sept | 21-Sept | 3-Oct | 1 day | 12 days* | 13 days* |
| Stratification Duration | Days | 119.4 | 127.4 | 138.6 | 8.0 | 11.2 | 19.2* |





Table 2.  Averaging periods for lake drivers used for calculating correlation coefficients of
each  pair of variables.

| Lake variables | Air temperature/wind speed | Secchi depth[1] |
|---|---|---|
| Ice-on date | November – December | 13 October – ice-on |
| Ice-off date | February – March | ice-on – ice-off |
| Max. ice thickness | January – March | ice-on – ice-off |
| Freeze-over water temperature | November – December | 13 October – ice-on |
| Date of stratification onset | April – May | ice-out – 10 May |
| Epilimnetic temperature | 16 July – 15 August | 30 June – 2 September |
| Hypolimnetic temperature | May | ice-out – 30 June |
| Hypolimnetic heating | July – August | 30 June – 2 September |
| Date of fall turnover | September | 3 September – 12 October |

[1]Six seasonal periods defined in Lathrop et al. (1996)



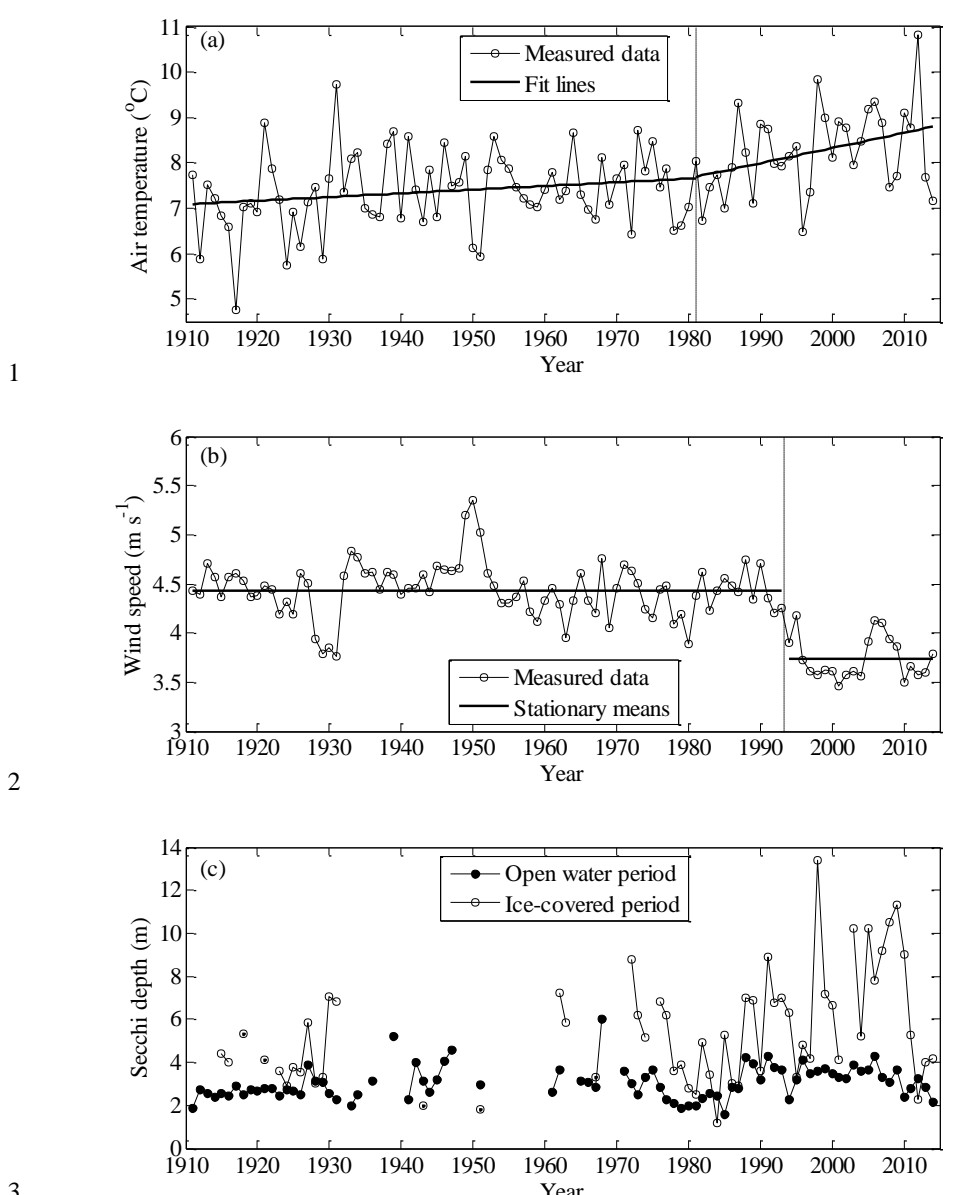

Figure 1: Historical record of annual average (a) air temperature and (b) wind speed in
Madison, WI, USA, and (c) Secchi depth in Lake Mendota.





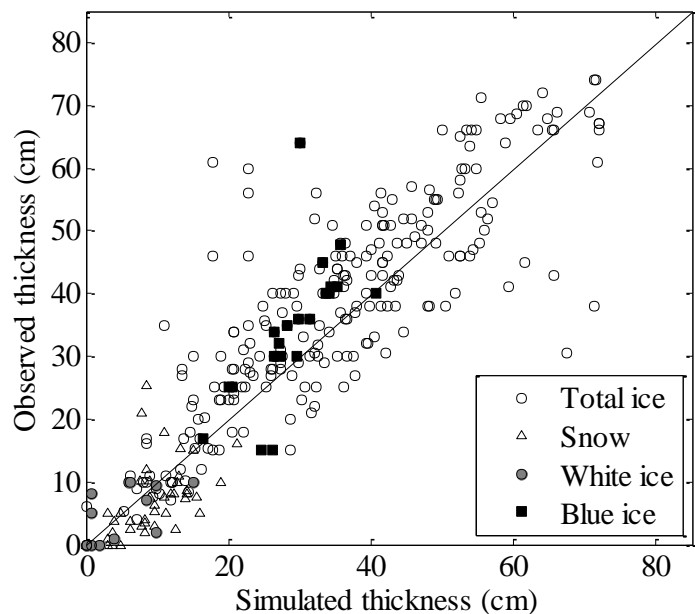

2      Figure 2. Comparison between observations and simulated total ice, snow cover, white ice,

3      and blue ice





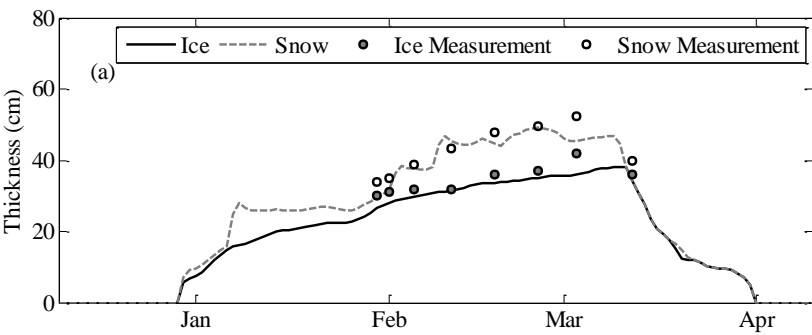

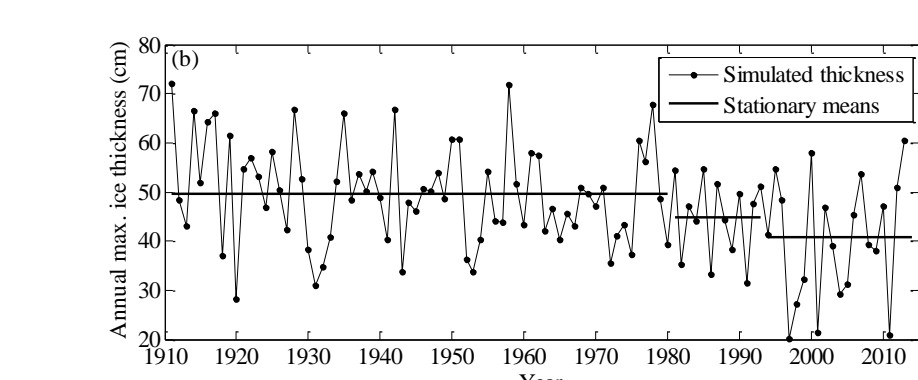

3    Figure 3: (a) simulated ice and snow with measured values for the winter 2009-2010 and (b)

4    simulated annual maximum total ice thickness.





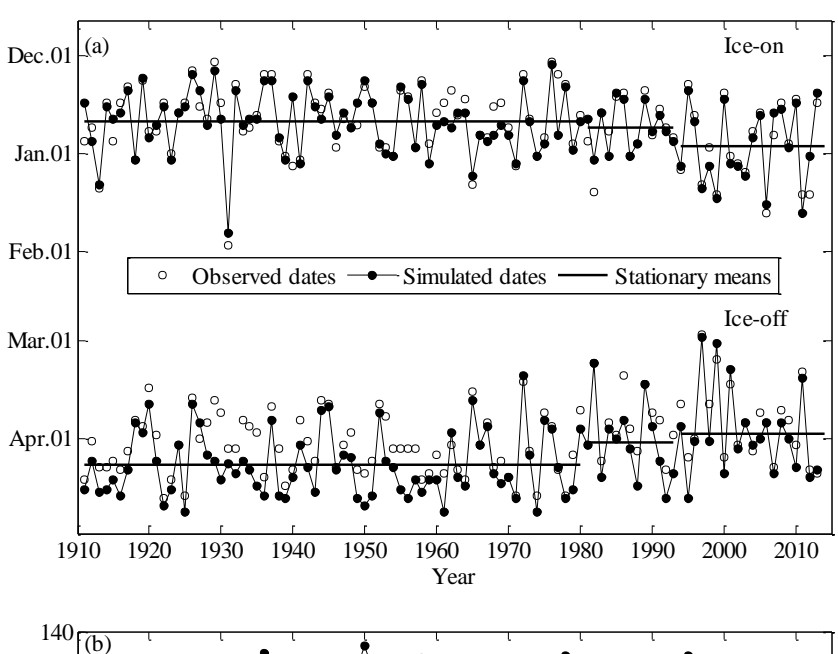

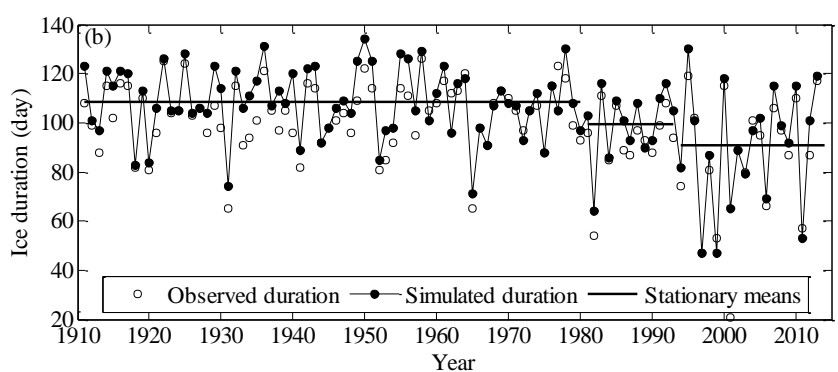

Figure 4: Observed and simulated (a) ice-on and ice-off dates and (b) ice cover duration. The
annual ice cover was plotted in the first year of the winter, e.g., the winter 1911-1912 was
plotted in the year 1911. Stationary means for three selected periods are denoted by solid
lines.





Figure 5: Time series of the simulated and observed volume-weighted near-surface (epilimnetic) temperature (0−10m) from 1911−2014.







Figure 6: Time series of the simulated and observed volume-weighted near-bottom (hypolimnetic) temperatures (20−25 m) from 1911−2014.





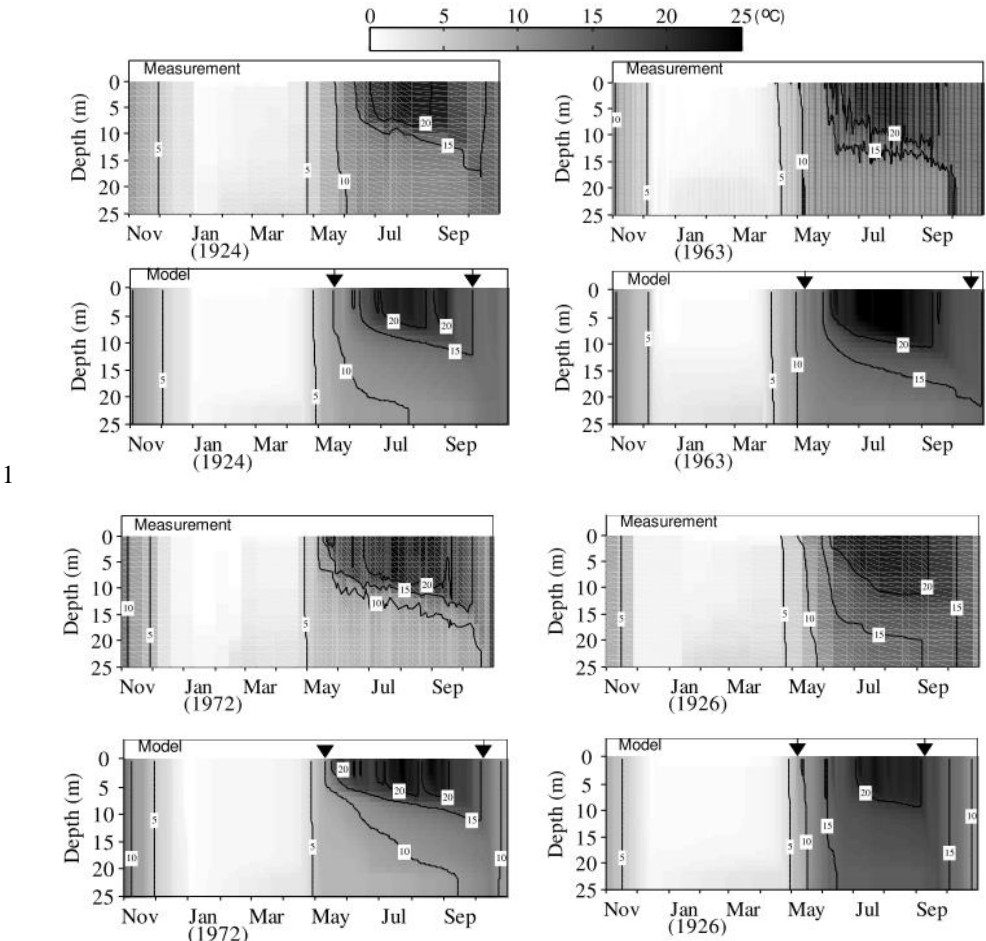

Figure 7: Comparison between the measured and simulated water temperatures as a function
of time and depth under the cold epilimnetic year (1924), warm epilimnetic year (1963), cold
hypolimnetic year (1972), and warm hypolimnetic year (1926) in Lake Mendota.  Arrows
mark the dates of onset of stratification and fall turnover.



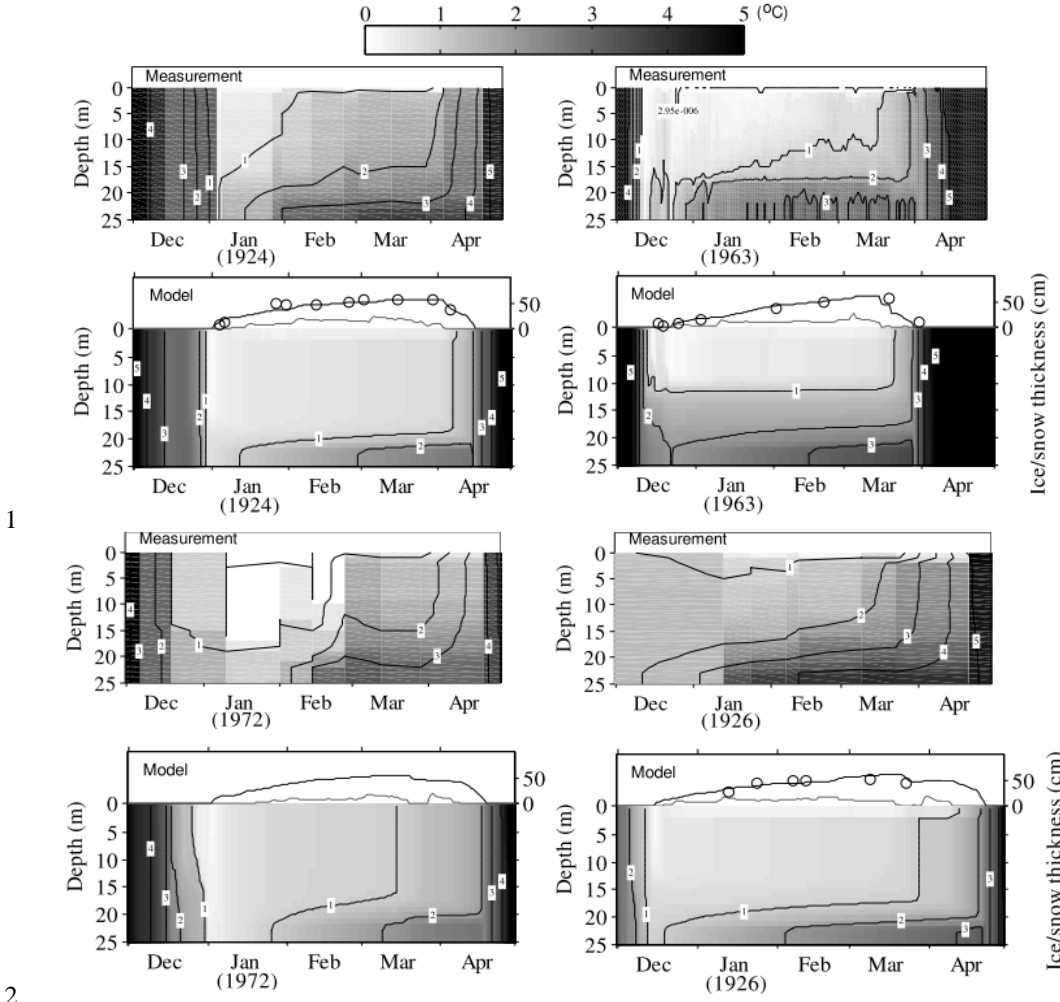

Figure 8: Comparison between the measured and simulated water temperatures and ice/snow
thickness under the cold epilimnetic year (1924), warm epilimnetic year (1963), cold
hypolimnetic year (1972), and warm hypolimnetic year (1926) in Lake Mendota during the
ice-covered period. Time series are simulated ice and snow thickness (solid lines) with
measurements (open circles).





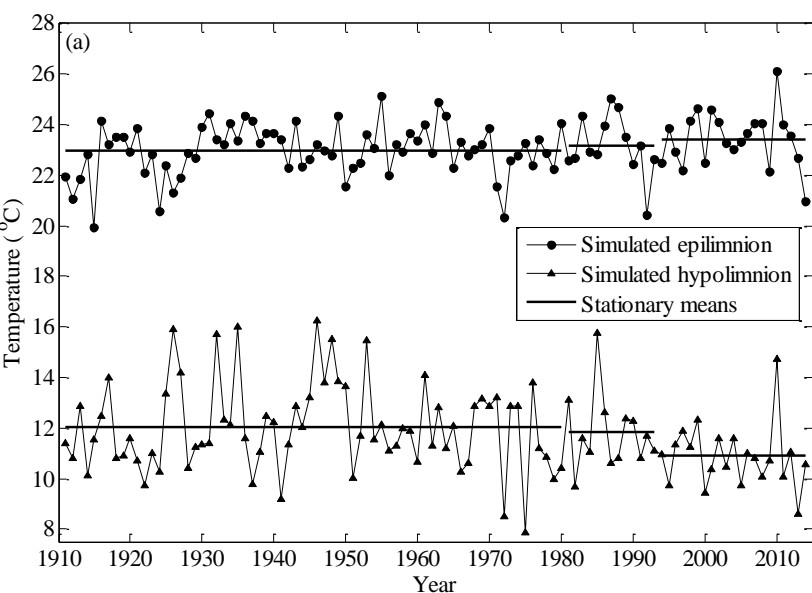

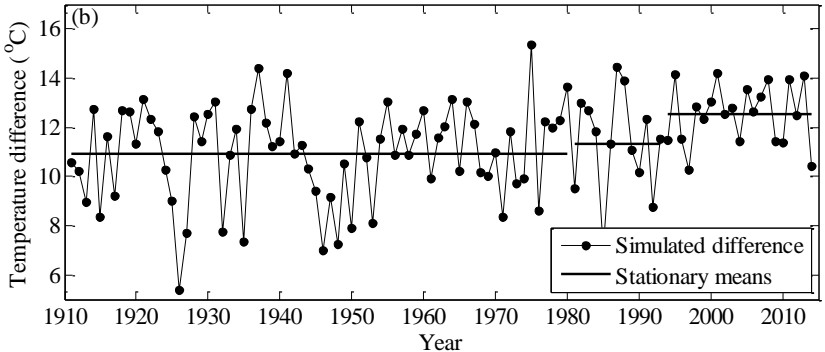

3  Figure 9: (a) Simulated mid-summer (16 July − 15 August) volume-averaged epilimnion and

4  hypolimnion temperatures, and (b) epilimnion-hypolimnion temperature difference.

5  Stationary means for three selected periods are denoted by solid lines





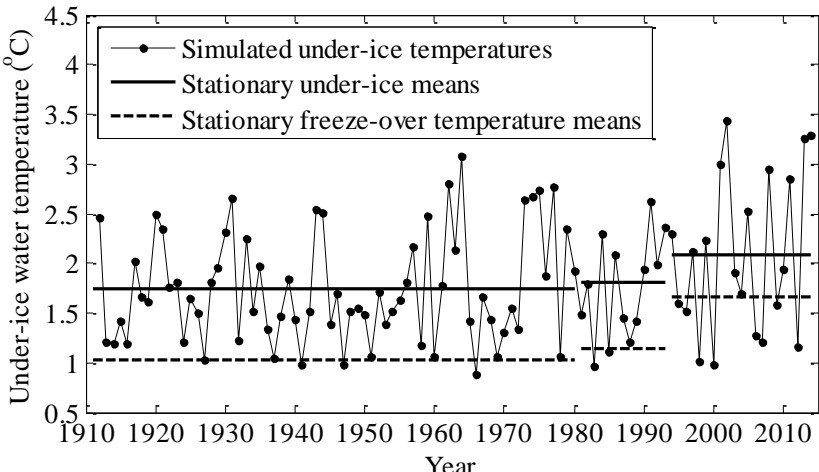

Figure 10: Simulated mean under-ice water temperatures (volume-weighted averages of all depths) during the ice-covered period between 1911 and 2014. The stationary means for three selected periods of under-ice water temperatures and freeze-over water temperatures are also plotted.





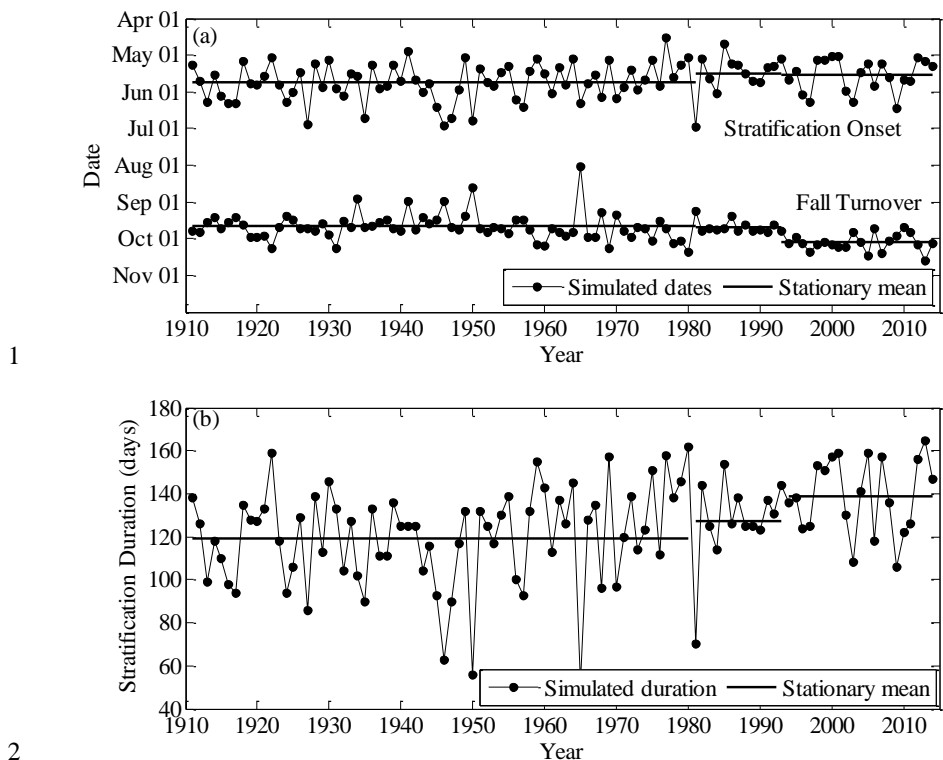

3   Figure 11: (a) Simulated date of stratification onset and fall turnover and (b) stratification

4   duration. Stationary means for three selected periods are denoted by solid lines.





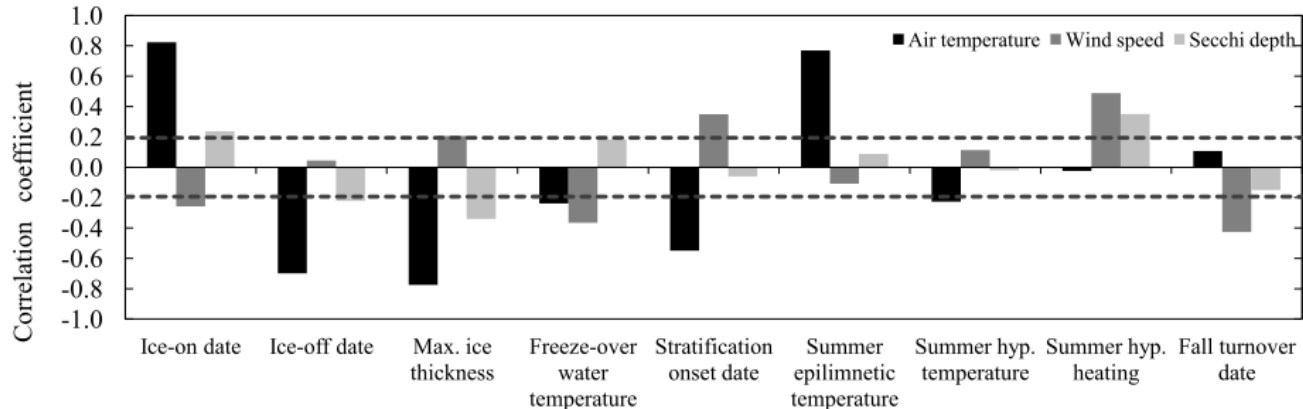

2    Figure 12: Correlation coefficients between lake variables and drivers. The critical value (dashed lines) for significant correlation (p<0.05)

3    0.193 (n = 104).