# Peer review of "Manuscript under review for journal Hydrol. Earth Syst. Sci."

_Hydrology and Earth System Sciences, 2015_

## Referee Comment (RC1) · R. Xia (Referee) · 31 Mar 2016

**Trends and Abrupt Changes in 104-years of Ice Cover and Water Temperature in a Dimictic Lake in Response to Air Temperature, Wind Speed, and Water Clarity Drivers**

**by M. R. Magee, C. H. Wu, D. M. Robertson, R. C. Lathrop, and D. P. Hamilton**

**Discussion by Renjie Xia**

General Comments

This article is well written, and the topic is interested.  Authors did an extensive literature review, and provided a large number of references to validate their work.  The conclusions presented in this article are useful.

Specific Comments

(1)  Both DYRESM (Dynamics Reservoir Simulation Model) and DYRESM-WQ (Dynamic Reservoir Simulation Model – Water Quality) developed by the Center for Water Research at the University of Western Australia have been extensively calibrated and verified through field work.  These models are reliable to use.

Authors added an ice and snow model to the DYRESM-WQ, and developed a new model called as DYRESM-WQ-I.  Authors wrote that this resulting model was validated and employed (validated using a long-term (1911-2014) observational dataset, then employed to simulate long-term (1911-2014) ice cover and water temperature in the lake).  One question has arisen what is the meaning of "validated" or "employed"?  In general, "calibrated" and "verified" are common used in scientific articles.  Authors should explain why using "validated" and "employed"?  In addition, seems this new model was validated and employed just once by using the same observational dataset.  Therefore, another question has arisen that the results obtained from this new model (validated and employed just once) is reliable?

(2)  One suggestion: dividing the long-term observational dataset to two groups, then using one for the "validated" purpose, and using another for the "employed" purpose.

(3)  Readers might be interested in the long-term 104-year continuous dataset and want to know how many variables observed are included in this dataset.  Summarizing a table to show all the observational variables in the dataset will be grateful to these readers.

---

## Referee Comment (RC2) · R. Zurek (Referee) · 5 Apr 2016

Scientific significance: Does the manuscript represent a substantial contribution to scientific progress within the scope of Hydrology and Earth System Sciences (substantial new concepts, ideas, methods, or data)? Yes Scientific quality: Are the scientific approach and applied methods valid? Are the results discussed in an appropriate and balanced way (consideration of related work, including appropriate references)? Yes Presentation quality: Are the scientific results and conclusions presented in a clear, concise, and well-structured way (number and quality of figures/tables, appropriate

use of English language)? Yes

1. Does the paper address relevant scientific questions within the scope of HESS? yes 2. Does the paper present novel concepts, ideas, tools, or data? Yes it concern concept and realisation 3. Are substantial conclusions reached? Yes 4. Are the scientific methods and assumptions valid and clearly outlined? Yes , very clear 5. Are the results sufficient to support the interpretations and conclusions? Yes 6. Is the description of experiments and calculations sufficiently complete and precise to allow their reproduction by fellow scientists (traceability of results)? Yes 7. Do the authors give proper credit to related work and clearly indicate their own new/original contribution Yes 8. Does the title clearly reflect the contents of the paper? Yes 9. Does the abstract provide a concise and complete summary? Yes 10. Is the overall presentation well structured and clear? Yes, perfect 11. Is the language fluent and precise? Yes 12. Are mathematical formulae, symbols, abbreviations, and units correctly defined and used? Yes 13. Should any parts of the paper (text, formulae, figures, tables) be clarified, reduced, combined, or eliminated? Generally is OK , however see comment 14. Are the number and quality of references appropriate? Yes 15. Is the amount and quality of supplementary material appropriate? Yes General comments.

To some extent, the discussion develops the chapter "results" and is focused on the examined lakes. In my subjective opinion lack of comparison with similar studies in lakes from another part of the world. I suggest to compare with European lakes with similar latitude For example

Skowron R. 2009. Changeability of the ice cover on the lakes of northern Poland in the light of climatic changes. Bull Geogr, 1,: 103–124 http://apcz.pl/czasopisma/index.php/BOGPGS/article/viewFile/2312/2296

Marszelewski W., Skowron R. 2006. Ice cover as an indicator of winter air temperature changes case study of the Polish lowland lakes. Hydrol. Sci. J. 41, 336-349 http://www.tandfonline.com/doi/pdf/10.1623/hysj.51.2.336

Choiński, A., L. Kolendowicz, J. Pociask-Karteczka, et al., 2010: Changes in lake ice cover on the Morskie Oko Lake in Poland (1971ïĂ▪2007). Adv. Clim. Change Res., 1, doi: 10.3724/SP.J.1248.2010.00071. Choiński A., Ptak M., Strzelczak A. 2013. Areal Variation In Ice Cover Thickness On Lake Morskie Oko (Tatra Mountains). Carpathian Journal of Earth and Environmental Sciences, 8, 3, 97 -102 https://www.researchgate.net/publication/263733557_Areal_variation_in_ice_cover_thickness_on_lake_morskie_oko_Tatr

Technical notes

Page 2, Line 11 insert space , 1994which Page 9 line 7 : correct Page 11, line 15. is: trend of .224, should be 0.334 Fig 2 I suggest to use filled triangle for snow, will be better visible Page 3, line 13 is Jiang et al. 2010, in references Jiang . . .. 2009 Page 3 line 32, Stefan et al 1996, lack in references Page 4 line 3 and 8 is Schindler et al 1996 lack in ref. Page 5 line 17, is Patterson 1981 lack in ref. Page 6 line 1 is McKay, 1968 in references lack year 1968 Page 8 line 10, Rodinov 2006 lack of year in references Page 8 line 16 is Kitchell 1992 im Litereture is Kitchell 2012 Page 9 line 20, is Lathrop et al 1996 in teferences is 1998 Page 18 line 3, is Lathrop et al. 1996, in literat. Is Lathrop et al 1998 Page 23 line 13 Stauffer and Armstrong 1986 m in references lack of year Page 23 kune 15 is Lee1973 insert space Page 23 line 18, is Rice 2015 lack in references ther is Rice et al 2014 Page 23 kine 23, is Carpanter et al 2007 lack in references Table 2 footnote Lathrop et al 1996 lack in references

Over-abound , in excess

Lathrop & Carpenter 2011

Not cited Malm et al 1997 not cited in the text. Rodionov 2005

Links to websites move to footnote

Remarks to figures

Fig. 2. use line 0.1-0.3 mm, not hairy, Snow symbol (triangle) fill.. Will be visible. Fig 4 .line use to open circles not hairy, minimum 0.1 to 0.3 mm

[Figure]

[Figure]

---

## Author Comment (AC1) · 12 Apr 2016

Reviewer 1: Dr. Renjie Xia The authors would like to offer sincere thanks to Dr, Renjie Xia for taking time to carefully review this manuscript and provide insightful comments to improve the quality of the manuscript. Below is a point-by-point response to issues raised in the manuscript.

General Comments This article is well written, and the topic is interested. Authors did an extensive literature review, and provided a large number of references to validate their work. The conclusions presented in this article are useful. We greatly thank

the reviewer for the compliment. Specific Comments (1) Both DYRESM (Dynamics Reservoir Simulation Model) and DYRESM-WQ (Dynamic Reservoir Simulation Model – Water Quality) developed by the Center for Water Research at the University of Western Australia have been extensively calibrated and verified through field work. These models are reliable to use. Authors added an ice and snow model to the DYRESM-WQ, and developed a new model called as DYRESM-WQ-I.

Authors wrote that this resulting model was validated and employed (validated using a long-term (1911-2014) observational dataset, then employed to simulate long-term (1911-2014) ice cover and water temperature in the lake). One question has arisen what is the meaning of "validated" or "employed"? In general, "calibrated" and "verified" are common used in scientific articles. Authors should explain why using "validated" and "employed"? In addition, seems this new model was validated and employed just once by using the same observational dataset. Therefore, another question has arisen that the results obtained from this new model (validated and employed just once) is reliable?

The authors thank the reviewer for pointing out this oversight and confusion in the manuscript. DYRESM-WQ-I was calibrated for Lake Mendota by setting the minimum layer thickness in the model. Other parameters for the hydrodynamic and ice models were chosen from previous literatures. Specifically, for the ice model, it is based heavily on the previously calibrated and validated Mixed Lake with Ice (MLI) model developed by Rogers et al (1995). Alterations to the model are for two-way coupling of the water-column dynamics to the ice model (MLI has only one-way coupling) and the addition of a time-varying sediment heat flux for all horizontal layers wherein the heat flux is dependent on both time-varying sediment temperatures and time-varying lake water temperatures. As the Rogers et al (1995) model has been previously validated through extensive field effort, we did not conduct further field-validation for this study. However, we calibrated the model for the period of 1995-2014. We add a new section, "2.2 Model calibration" and validated compared to observed data for the full simulation period 1911-2014. We have added the following text to improve clarity on model development, calibration, and validation.

P 5, L18-20: " The ice model is based upon the MLI model of Rogers et al., (1995) with alterations to two-way coupling of the water-column dynamics to the ice model and the addition of time-dependent sediment heat flux for all horizontal layers."

P7, L21 – P8, L2: "The model was calibrated for the period 1995-2014 by varying the minimum layer thickness over values ranging from 0.05 m to 0.5 m at 0.025 m intervals. Layer thickness values were evaluated for the least amount of deviation between predicted and observed temperature values for Lake Mendota over the period. Based on this analysis, a minimum layer thickness of 0.125 m was chosen as the best setting to predict water temperature at all depths. Other parameter values in the hydrodynamic and ice cover models were obtained from literature values (Table 1). To evaluate the performance of the model, root-mean square error (RMSE) was used to compare simulated and observed ice cover and water temperature values for the full model period (1911-2014; see Sect. 4.2). Simulated and observed values are compared directly, with the exception of aggregation of water temperature measurements to daily intervals where sub-daily intervals were available."

Additionally, a new table, Table 1, has been added to the manuscript to provide parameter values used in the hydrodynamic and ice model portions of DYRESM-WQ-I.

(2) One suggestion: dividing the long-term observational dataset to two groups, then using one for the "validated" purpose, and using another for the "employed" purpose. Please see the response to comment 1. Specifically, we calibrated the model for the period of 1995-2014. Afterwards, we validated the model for the full simulation period 1911-2014.

(3) Readers might be interested in the long-term 104-year continuous dataset and want to know how many variables observed are included in this dataset. Summarizing a table to show all the observational variables in the dataset will be grateful to these

readers.

The authors agree that the observation datasets used for model input and calibration/validation are valuable to readers. Indeed the variables are listed in the subtitles of section 3. Including another table listing datasets in addition to what is included within the text may be repetitive. Instead, we have revised sections of the text to further detail where raw datasets are available and where data adjustments were made to improves the clarity of datasets.

---

## Author Comment (AC2) · 12 Apr 2016

Reviewer 2: Dr. Roman Zurek The authors would like to thank Dr. Roman Zurek for carefully reading the manuscript and providing thoughtful and helpful comments. We have revised the manuscript accordingly and detailed these changes in the point-by-point response below.

Scientific significance: Does the manuscript represent a substantial contribution to scientific progress within the scope of Hydrology and Earth System Sciences (substantial new concepts, ideas, methods, or data)? Yes Scientific quality: Are

the scientific approach and applied methods valid? Are the results discussed in an appropriate and balanced way (consideration of related work, including appropriate references)? Yes Presentation quality: Are the scientific results and conclusions presented in a clear, concise, and well-structured way (number and quality of figures/tables, appropriate use of English language)? Yes 1. Does the paper address relevant scientific questions within the scope of HESS? yes 2. Does the paper present novel concepts, ideas, tools, or data? Yes it concern concept and realization 3. Are substantial conclusions reached? Yes 4. Are the scientific methods and assumptions valid and clearly outlined? Yes, very clear 5. Are the results sufficient to support the interpretations and conclusions? Yes 6. Is the description of experiments and calculations sufficiently complete and precise to allow their reproduction by fellow scientists (traceability of results)? Yes 7. Do the authors give proper credit to related work and clearly indicate their own new/original contribution Yes 8. Does the title clearly reflect the contents of the paper? Yes 9. Does the abstract provide a concise and complete summary? Yes 10. Is the overall presentation well structured and clear? Yes, perfect 11. Is the language fluent and precise? Yes 12. Are mathematical formulae, symbols, abbreviations, and units correctly defined and used? Yes 13. Should any parts of the paper (text, formulae, figures, tables) be clarified, reduced, combined, or eliminated? Generally is OK, however see comment 14. Are the number and quality of references appropriate? Yes 15. Is the amount and quality of supplementary material appropriate? Yes

We appreciate the positive comments assessed by Dr. Roman Zurek. We address the specific comments raised by the reviewer to improve the quality of the manuscript.

General comments. To some extent, the discussion develops the chapter "results" and is focused on the examined lakes. In my subjective opinion lack of comparison with similar studies in lakes from another part of the world. I suggest to compare with European lakes with similar latitude For example: Skowron R. 2009. Changeability of the ice cover on the lakes of north- ern Poland in the light of climatic changes. Bull Geogr,

1,: 103–124 http://apcz.pl/czasopisma/index.php/BOGPGS/article/viewFile/2312/2296

Marszelewski W., Skowron R. 2006. Ice cover as an indicator of winter air temperature changes case study of the Polish lowland lakes. Hydrol. Sci. J. 41, 336-349 http://www.tandfonline.com/doi/pdf/10.1623/hysj.51.2.336

Choi′nski, A., L. Kolendowicz, J. Pociask-Karteczka, et al., 2010: Changes in lake ice cover on the Morskie Oko Lake in Poland (1971ï ËŸA 2007). Adv. Clim. Change Res., 1, doi: 10.3724/SP.J.1248.2010.00071.

Choi′nski A., Ptak M., Strzelczak A. 2013. Areal Variation In Ice Cover Thickness On Lake Morskie Oko (Tatra Mountains). Carpathian Journal of Earth and Environmental Sciences, 8, 3, 97 -102 https://www.researchgate.net/publication/263733557_Areal_variation_in_ice_cover_thickness_on_lake_morskie_oko_Tatr

We have included these references, of which we were previously unaware. Many thanks for providing the references. We have incorporated comparisons to the earlier studies in European lakes where appropriate in the text.

Pg 14, L7-10: "These results are much smaller than those for European lakes of similar latitudes (Choiński et al., 2010, 2013; Marszelewski and Skowron, 2006; Skowron, 2009), with changes ranges from 0.20 to 0.60 cm yr-1, almost double that of Lake Mendota if the current change per year is extended to change per century."

Pg 15, L8-16: "Similar tendencies have been observed at other lakes, which show decreasing ice cover duration from later ice on dates and earlier ice off dates (Choiński et al., 2010, 2013; Marszelewski and Skowron, 2006; Skowron, 2009). However, lakes in near the Great Lakes, North America and Poland have shown larger rates of change over periods of less than a century. For example, Jensen et al. (2007) observed average ice duration decreases of 5.3 days decade-1 from 1975-2004 in the Great Lakes Region, and Polish lakes had observed changes as large as 0.8 to 0.9 days year-1 for the peirod 1961-2000 (Marszelewski and Skowron, 2006) and 0.5 to 0.6

days year-1 from 1956-2005 (Skowron, 2009)."

Pg 23, L14-16: "Similarly, lakes in Poland show a considerable statistical relationship between ice cover and the North Atlantic Oscillation winter indexes (Skowron, 2009), indicating that ice cover may be driven by other large oscillations as well."

Technical notes Page 2, Line 11 insert space , 1994 which This change has been made

Page 9 line 7 : correct The spacing issue has been corrected.

Page 11, line 15. is: trend of .224, should be 0.334 We have corrected this on Page 12 – Line 11

Fig 2 I suggest to use fi̧lled triangle for snow, will be better visible We have made this change in Figure 2 as suggested.

Page 3, line 13 is Jiang et al. 2010 We change to Jiang et al. 2009 Page 3 line 32, Stefan et al 1996, lack in references Reference has been added on Page 35 Line 10-13.

Page 4 line 3 and 8 is Schindler et al 1996 lack in ref. Reference has been added on Page 34 Line 27-30.

Page 5 line 17, is Patterson 1981 lack in ref. Reference has been added on Page 33 Line 13-14.

Page 6 line 1 is McKay, 1968 in references lack year 1968 Reference has been added on Page 32 Line 37-38.

Page 8 line 10, Rodinov 2006 lack of year in references Reference has been added on Page 34 Line 9-10.

Page 8 line 16 is Kitchell 1992 im Litereture is Kitchell 2012 The year has been changed to 2012.

Page 9 line 20, is Lathrop et al 1996 in teferences is 1998 The reference has been

changed to 1996. See page 32 Line 6

Page 18 line 3, is Lathrop et al. 1996, in literat. Is Lathrop et al 1998 The reference has been changed to 1996. See page 32 Line 6

Page 23 line 13 Stauffer and Armstrong 1986 m in references lack of year The year has been added.

Page 23 kune 15 is Lee1973 insert space Inert the space

Page 23 line 18, is Rice 2015 lack in references ther is Rice et al 2014 Change to 2015 see Page 33, Line 32

Page 23 kine 23, is Carpanter et al 2007 lack in references It should be Carpenter et al. 1992.

Table 2 footnote Lathrop et al 1996 lack in references The reference has been added on Page 32 Line 4-6.

Over-abound , in excess Revised.

Lathrop & Carpenter 2011Not cited, Malm et al 1997 not cited in the text, Rodionov 2005 All are removed from the references.

Thank you to the reviewer for pointing out these errors in references and citations. All the above errors in references have been corrected accordingly. Furthermore, we have reviewed the reference list carefully to update other missing or improperly formatted references in the text and reference list.

Links to websites move to footnote The author guidelines for this journal specify that the use of footnotes should be avoided as much as possible. As result, we keep them in the text in an effort to conform to the journal guidelines.

Remarks to figures Fig. 2. use line 0.1-0.3 mm, not hairy, Snow symbol (triangle) fill.. Will be visible. We have changed the fill on the triangle representing snow

thickness to make the symbols more visible.

Fig 4.line use to open circles not hairy, minimum 0.1 to 0.3 mm We have changed the thickness of the lines on the charts to make them more legible.

---

## Author Comment (AC3) · 12 Apr 2016

[revised manuscript text omitted]
 the MLI model of Rogers et al., (1995) with alterations to two-way coupling of the water-column dynamics to the ice model and the addition of time-dependent sediment heat flux for all horizontal layers. Ice growth in the model is based upon a quasi-steady state assumption that the time scale for heat conduction through the ice is short relative to the time scale of meteorological forcing (Patterson and Hamblin, 1988; Rogers et al., 1995). This assumption is valid under a Stefan Number <0.1 (Hill and Kucera, 1983). The ice module is applied when the simulated surface water temperature first drops below 0 $^{\circ}$C; the initial ice thickness is set to a value of 5 cm to address effects of partial ice cover (Patterson and Hamblin, 1988; Vavrus et al., 1996). Upward conductive heat flux between ice/snow cover and the atmosphere, $q_0$, is determined by numerically solving the quasi-steady state heat conduction equations (Rogers et al., 1995) and assigning appropriate boundary conditions to the water, ice, and atmospheric interfaces. At the ice (or snow) surface, a heat flux balance provides the condition for surface melting, and accretion or ablation of ice is determined through the heat flux at the ice-water interface. Imbalance between heat conduction through ice and the heat flux from the water to the ice gives the rate of change of ice thickness at the ice-water interface. Snow conductivity is estimated from its density using an empirical equation (Ashton, 1986), and snow compaction is based on an exponential decay formula (McKay, 1968), with snow compaction parameters based on air temperature and   snowfall/rainfall (Rogers et al., 1995). Snow (white) ice is generated in response to flooding, when the mass of snow that can be supported by the ice cover is exceeded. The uppermost solid layer of ice or snow is adjusted in thickness at the end of each 1-hr model time step according to the balance of the heat budget. When ice thickness decreases to less than 5 cm, open water conditions are restored.

Sediment heat flux, the main external source of heat after freezing, is important to water temperatures beneath ice cover (Ellis et al., 1991). Sediment heat flux (Fang and Stefan, 1996a) is included as a source/sink term for each Lagrangian layer to closely simulate under-ice water temperatures. A simple diffusion relation (Rogers et al., 1995) is used to estimate heat transfer from the sediments to the water column, $q_{sed}$:

$$q_{sed} = K_{sed} \frac{dT}{dz}. \tag{1}$$

where $K_{sed}$ is the sediment conductivity (=1.2 W m$^{-1}$ $^{o}$C$^{-1}$). d$T$/d$z$, the temperature gradient across the sediment-water interface, is estimated as:

$$\frac{dT}{dz} \approx \frac{T_s - T_w}{z_{sed}}. \tag{2}$$

where $T_s$ is the sediment temperature, $T_w$ is the water temperature adjacent to the sediment surface, which varies hourly and with depth, and $z_{sed}$ is the distance beneath the water-sediment interface at which the sediment temperature becomes largely invariant. From data collected at four locations on Lake Mendota (Birge et al., 1927), it was found that sediment temperatures varied little at 5 m depth below the sediment-water interface, so $z_{sed}$ is set to be 5 m. In addition, data from Birge et al., (1927) is used to fit a curve to describe the seasonal variation of $T_s$

$$T_s = 9.7 + 2.7 \sin\left[\frac{2\pi(D-151)}{TD}\right] \tag{3}$$

where D is the number of days from the start of the year and TD is the total number of days for the year of interest (365 or 366). The vertical transfer of heat in the water column beneath the ice is regulated by an assigned thermal diffusion coefficient. The formulation is the same as that originally used in DYRESM-WQ (Hamilton and Schladow, 1997) to simulate heat transfer throughout the open water period. This produced diffusivities that were within the range of measurements by Ellis et al., (1991) of 1-3 times greater than molecular values.

Input for the model includes lake morphometry (lake volume and surface area as a function of elevation), initial vertical profiles for water temperature and salinity, Secchi depth, meteorological variables, and inflows/outflows. The model calculates the surface heat fluxes using meteorological variables: total daily shortwave radiation, daily cloud cover, air vapor pressure, daily average wind speed, air temperature, and precipitation. During the simulation, all parameters/coefficients in the model are kept constant. The time step in the model for calculating water temperature, water budget, and ice thickness was set to 1 hr. Snow ice compaction, and snowfall and rainfall components are updated at a daily time step, corresponding to the frequency of meteorological data input. Cloud cover, air pressure, wind speed, and air temperature are assumed to be constant throughout the day, and precipitation is assumed uniformly distributed. Shortwave radiation distribution throughout the day is computed based on the lake latitude and the day or year. The DYRESM-WQ-I model is calibrated using measured lake variables including water level, temperature profiles, ice thickness, and ice on and ice off dates. The overall simulation period was 104 years, starting with an isothermal (measured) water column temperature of 3.1 °C on 7 April 1911 and ending on 31 October 2014.

**2.2        Model calibration**

The model was calibrated for the period 1995-2014 by varying the minimum layer thickness over values ranging from 0.05 m to 0.5 m at 0.025 m intervals. Layer thickness values were evaluated for the least amount of deviation between predicted and observed temperature values for Lake Mendota over the period. Based on this analysis, a minimum layer thickness of 0.125 m was chosen as the best setting to predict water temperature at all depths. Other parameter values in the hydrodynamic and ice cover models were obtained from literature values (Table 1). To evaluate the performance of the model, root-mean square error (RMSE) was used to compare simulated and observed ice cover and water temperature values for the full model period (1911-2014; see Sect. 4.2). Simulated and observed values are compared directly, with the exception of aggregation of water temperature measurements to daily intervals where sub-daily intervals were available.

**2.3 Piecewise regression algorithm**

Breakpoints in the air temperature trend over the study period were determined using a piecewise linear regression (PLR) method (Tomé and Miranda, 2004; Toms and Lesperance, 2003; Ying et al., 2015) that assumes continuity in the trendlines across the breakpoint. The piecewise linear regression model finds the breakpoint, B, that minimizes the residual sum of squares ($RSS_{PLR}$) of the model between the two phases (Ying et al., 2015)

$$RSS_{PLR} = \sum_{i=1}^{n} \left[ y_i - a - k_1 x_i - \left( k_2 - k_1 \right) \max \left( x_i - B, 0 \right) \right]^2 \tag{4}$$

[revised manuscript text omitted]

### 3.4 River inflow and outflow

Daily inflow measurements have been made on selected Lake Mendota tributaries since 1974, and daily outflow from the lake has been measured since 1975. Streamflow for the tributaries were obtained from: http://waterdata.usgs.gov/wi/nwis/sw/. Daily outflow from 1975 to 1997

were calculated from gate/lock/by-pass pipe USGS ratings and local governmental daily operational records. Daily out from 1998 to 2003 were estimated from monthly outflow estimates from downstream Lake Waubesa and Mendota outlet data from earlier years. Daily outflow from 2003 to the present, were obtained from http://waterdata.usgs.gov/wi/nwis/uv/?site_no=05428000. 
[revised manuscript text omitted]

similar latitudes (Choiński et al., 2010, 2013; Marszelewski and Skowron, 2006; Skowron,
2009), with changes ranges from 0.20 to 0.60 cm yr$^{-1}$, almost double that of Lake Mendota if
the current change per year is extended to change per century. A t-test of the mean values
shows a statistically significant ($p<0.05$) difference in the mean annual maximum ice
thickness between period 1 (1911−1980) and period 3 (1994−2014). Period 2 was not
statistically different from either of the other two periods (Table 2).

**4.2.2 Ice-on and ice-off dates**

Figure 4 shows measured and simulated ice-on date, ice-off date, and ice duration on the lake.
The measured ice-on date is defined as the first day when the lake becomes fully ice-covered,
and the ice-off date is the last day of the ice breakup before the open water season. Simulated
results and observations are in good agreement, with a mean absolute difference of 2.3 days
(RMSE =2.4 days) for ice-on date and 5.7 days (RMSE = 4.8 days) for ice-off date. It is noted
that both the mean error and RMSE are much smaller than the interannual variability in the
observed ice-on dates (standard deviation SD = 11.2 days) and ice-off dates (SD = 10.6 days).
Interannual variations in ice cover on Lake Mendota were large in the past century. Observed
ice-on dates ranged from 3 December (1929) to 30 January (1931): a range of 58 days. Ice-off
dates ranged from 27 February (1998) to 20 April (1923): a range of 51 days. The model
successfully captures the interannual variations of ice-on and ice-off dates. For example, it
reproduces the unusual late ice-on date in 1930-1931, and several noticeable early ice
breakups associated with intense El Niño -Southern Oscillation (ENSO) events, i.e., 1965,
1972, 1982, and 1997 (Anderson et al., 1996; Magnuson et al., 2000; Robertson et al., 2002)
Figure 4b shows the ice duration, defined as the period between ice-on and ice-off dates. The
mean absolute difference in ice duration between the model results and observations is 6.6
days (RMSE = 6.1 days), compared to the observed SD of 17.9 days. Overall, we consider
the model performs well in simulating ice on and off dates and ice cover duration.

Long-term trends in ice formation are examined by applying linear regression to the model results and observed data. Figure 4 clearly shows progressively later freezing, earlier breakup, and shorter duration in Lake Mendota from 1911 to 2014. Based on model results, ice-on dates became later by 9.0 days per century, ice-off dates became earlier by 12.3 days per century, and ice duration shortened by 21.3 days per century. Model results are in good agreement with those obtained from the observed data (7.4 days later ice-on dates, 9.3 days earlier for ice-off dates, and 18.0 days shorter duration per century). All linear trends from the observed and simulated data are statistically significant ($p<0.05$). Similar tendencies have been observed at other lakes, which show decreasing ice cover duration from later ice on dates and earlier ice off dates (Choiński et al., 2010, 2013; Marszelewski and Skowron, 2006; Skowron, 2009). However, lakes in near the Great Lakes, North America and Poland have shown larger rates of change over periods of less than a century. For example, Jensen et al. (2007) observed average ice duration decreases of 5.3 days decade[-1] from 1975-2004 in the Great Lakes Region, and Polish lakes had observed changes as large as 0.8 to 0.9 days year[-1] for the peirod 1961-2000 (Marszelewski and Skowron, 2006) and 0.5 to 0.6 days year[-1] from 1956-2005 (Skowron, 2009). Mean values of ice-on, ice-off, and ice duration for the three selected periods, i.e., 1911–1980, 1981–1993, and 1994–2010, (see Table 2 and Fig. 4) show a statistically significant ($p<0.05$, based upon t-test) difference between period 1 (1911−1980) and period 3 (1994−2014) for all ice cover variables and a statistically significant ($p<0.05$) difference between period 1 and period 2 (1981−1993) for ice cover duration.

**4.2.3 Water temperature**

The performance of the model in simulating water temperatures in the lake is presented in Figures 5 and 6. For temperatures at the near-surface, we use the simulated volume-weighted mean water temperatures between depths of 0 and 10m. The simulated epilimnetic temperatures compare well with those estimated from the measured data; the annual mean absolute error is 0.69 °C (n=3,239; RMSE=0.30 °C), demonstrating the ability of the model to simulate the heat budget in the upper mixed layer of the lake. Near-surface temperatures range from about 0 °C in winter to 26.1 °C in summer, and strongly respond to the net surface heat flux, as illustrated by the 2–3 °C variations that occur in both simulated and measured data, e.g. in the summer of 1921, 1960, 1994, and 2009 (Fig. 5). Overall the model captures interannual variations in epilimnetic temperatures even during extreme years; for example, the relatively high maximum epilimnetic temperatures (> 25 °C) in the summers of 1949, 1955,

1963, 1987, 1988, and 1999, and relatively cold annual maximum epilimnetic temperatures (~

22 $^{o}$C) in the summers of 1915, 1924, and 1942.

Figure 6 shows the comparison between the simulated volume-weighted near-bottom (within the hypolimnion) temperatures over the depths between 20−25 m and those estimated from the measured data. The mean absolute error is 1.04 $^{o}$C (n=3,239; RMSE=0.53 $^{o}$C). Near- bottom temperatures range from 0.2$^{o}$C in winter to 19.1$^{o}$C in autumn, which is approximately

[revised manuscript text omitted]

http://link.springer.com/chapter/10.1007/978-1-4612-0695-8_11 (Accessed 1 October 2015), 1998.

Choiński, A., Kolendowicz, L., Pociask-Karteczka, J. and Sobkowiak, L.: Changes in Lake Ice Cover on the Morskie Oko Lake in Poland (1971–2007), Adv. Clim. Change Res., 1(2), 71–75, doi:10.3724/SP.J.1248.2010.00071, 2010.

Choiński, A., Ptak, M. and Strzelczak: Areal variation in ice cover thickness on lake morskie oko (Tatra mountains), Carpathian J. Earth Environ. Sci., 8(3), 97–102, 2013.

Desai, A. R., Austin, J. A., Bennington, V. and McKinley, G. A.: Stronger winds over a large lake in response to weakening air-to-lake temperature gradient, Nat. Geosci., 2(12), 855–858, doi:10.1038/ngeo693, 2009.

De Stasio, B. T., Hill, D. K., Kleinhans, J. M., Nibbelink, N. P. and Magnuson, J. J.: Potential effects of global climate change on small north-temperate lakes: Physics, fish, and plankton, Limnol. Oceanogr., 41(5), 1136–1149, doi:10.4319/lo.1996.41.5.1136, 1996.

Dobiesz, N. E. and Lester, N. P.: Changes in mid-summer water temperature and clarity across the Great Lakes between 1968 and 2002, J. Gt. Lakes Res., 35(3), 371–384, doi:10.1016/j.jglr.2009.05.002, 2009.

Duguay, C. R., Flato, G. M., Jeffries, M. O., Ménard, P., Morris, K. and Rouse, W. R.: Ice-cover variability on shallow lakes at high latitudes: model simulations and observations, Hydrol. Process., 17(17), 3465–3483, doi:10.1002/hyp.1394, 2003.

Elliot, A. J., Thackeray, S. J., Huntingford, C. and Jones, R. G.: Combining a regional climate model with a phytoplankton community model to predict future changes in phytoplankton in lakes, Freshw. Biol., 50(8), 1404–1411, doi:10.1111/j.1365-2427.2005.01409.x, 2005.

Ellis, C. R., Stefan, H. G. and Gu, R.: Water Temperature Dynamics and Heat Transfer Beneath the Ice Cover of a Lake, Limnol. Oceanogr., 36(2), 324–335, 1991.

Fang, X. and Stefan, H. G.: Dynamics of heat exchange between sediment and water in a lake, Water Resour. Res., 32(6), 1719–1727, doi:10.1029/96WR00274, 1996a.

Fang, X. and Stefan, H. G.: Long-term lake water temperature and ice cover simulations/measurements, Cold Reg. Sci. Technol., 24(3), 289–304, doi:10.1016/0165-232X(95)00019-8, 1996b.

Fang, X. and Stefan, H. G.: Simulated climate change effects on dissolved oxygen characteristics in ice-covered lakes, Ecol. Model., 103(2–3), 209–229, doi:10.1016/S0304-3800(97)00086-0, 1997.

Fang, X. and Stefan, H. G.: Simulations of climate effects on water temperature, dissolved oxygen, and ice and snow covers in lakes of the contiguous U.S. under past and future climate scenarios, Limnol. Oceanogr., 54(6part2), 2359–2370, doi:10.4319/lo.2009.54.6_part_2.2359, 2009.

Fee, E. J., Hecky, R. E., Regehr, G. W., Hendzel, L. L. and Wilkinson, P.: Effects of Lake
Size on Nutrient Availability in the Mixed Layer during Summer Stratification, Can. J. Fish.
Aquat. Sci., 51(12), 2756–2768, doi:10.1139/f94-276, 1994.

Fee, E. J., Hecky, R. E., Kasian, S. E. M. and Cruikshank, D. R.: Effects of lake size, water
clarity, and climatic variability on mixing depths in Canadian Shield lakes, Limnol.
Oceanogr., 41(5), 912–920, doi:10.4319/lo.1996.41.5.0912, 1996.

Findlay, D. L., Kasian, S. E. M., Stainton, M. P., Beaty, K. and Lyng, M.: Climatic influences
on algal populations of boreal forest lakes in the Experimental Lakes Area, Limnol.
Oceanogr., 46(7), 1784–1793, doi:10.4319/lo.2001.46.7.1784, 2001.

Francis, T. B., Wolkovich, E. M., Scheuerell, M. D., Katz, S. L., Holmes, E. E. and Hampton,
S. E.: Shifting Regimes and Changing Interactions in the Lake Washington, U.S.A., Plankton
Community from 1962–1994, PLoS ONE, 9(10), e110363,
doi:10.1371/journal.pone.0110363, 2014.

Fu, G., Charles, S. P. and Yu, J.: A critical overview of pan evaporation trends over the last
50 years, Clim. Change, 97(1-2), 193–214, doi:10.1007/s10584-009-9579-1, 2009.

Gao, S. and Stefan, H. G.: Multiple Linear Regression for Lake Ice and Lake Temperature
Characteristics, J. Cold Reg. Eng., 13(2), 59–77, doi:10.1061/(ASCE)0887-
381X(1999)13:2(59), 1999.

Gao, S. and Stefan, H. G.: Potential Climate Change Effects on Ice Covers of Five Freshwater
Lakes, J. Hydrol. Eng., 9(3), 226–234, doi:10.1061/(ASCE)1084-0699(2004)9:3(226), 2004.

Gunn, J. M.: Impact of the 1998 El Niño event on a Lake Charr, Salvelinus Namaycush,
Population Recovering from Acidification, Environ. Biol. Fishes, 64(1-3), 343–351,
doi:10.1023/A:1016058606770, 2002.

Hamilton, D. P. and Schladow, S. G.: Prediction of water quality in lakes and reservoirs. Part
I — Model description, Ecol. Model., 96(1–3), 91–110, doi:10.1016/S0304-3800(96)00062-2,
1997.

Hartmann, D. L., Klein Tank, A. M. G., Rusticucci, M., Alexander, L. V., Brönnimann, S.,
Charabi, Y., Dentener, F. J., Dlugokencky, E. J., Easterling, D. R., Kaplan, A., Soden, B. J.,
Thorne, P. W., Wild, M. and Zhai, P. M.: Climate change 2013: the physical science basis:
Working Group I contribution to the Fifth assessment report of the Intergovernmental Panel
on Climate Change, edited by T. Stocker, D. Qin, G.-K. Plattner, M. Tignor, S. K. Allen, J.
Boschung, A. Nauels, Y. Xia, V. Bex, and P. M. Midgley, Cambridge University Press, New
York., 2013.

Heino, J., Virkkala, R. and Toivonen, H.: Climate change and freshwater biodiversity:
detected patterns, future trends and adaptations in northern regions, Biol. Rev., 84(1), 39–54,
doi:10.1111/j.1469-185X.2008.00060.x, 2009.

Hill, J. M. and Kucera, A.: Freezing a saturated liquid inside a sphere, Int. J. Heat Mass
Transf., 26(11), 1631–1637, doi:10.1016/S0017-9310(83)80083-0, 1983.

Hocking, G. C. and Straškraba, M.: The Effect of Light Extinction on Thermal Stratification in Reservoirs and Lakes, Int. Rev. Hydrobiol., 84(6), 535–556, doi:10.1002/iroh.199900046, 1999.

Hsieh, Y.: Modeling ice cover and water temperature of Lake Mendota, PhD Thesis, University of Wisconsin-Madison, Madison, Wisconsin, USA., 2012.

Huber, V., Adrian, R. and Gerten, D.: Phytoplankton response to climate warming modified by trophic state, Limnol. Oceanogr., 53(1), 1–13, doi:10.4319/lo.2008.53.1.0001, 2008.

Imberger, J. and Patterson, J. C.: Dynamic reservoir simulation model - DYRESM: 5, in Transport Models for Inland and Coastal Waters, edited by H. B. Fischer, pp. 310–361, Academic Press., 1981.

IPCC: Summary for Policymakers, in Climate Change 2013: The Physical Science Basis. Contribution of Working Group I to the Fifth Assessment Report of the Intergovernmental Panel on Climate Change, edited by T. Stocker, D. Qin, G.-K. Plattner, M. Tignor, S. K. Allen, J. Boschung, A. Nauels, Y. Xia, V. Bex, and P. M. Midgley, Cambridge University Press, New York, NY, USA., 2013.

Jensen, O. P., Benson, B. J., Magnuson, J. J., Card, V. M., Futter, M. N., Soranno, P. A. and Stewart, K. M.: Spatial analysis of ice phenology trends across the Laurentian Great Lakes region during a recent warming period, Limnol. Oceanogr., 52(5), 2013–2026, doi:10.4319/lo.2007.52.5.2013, 2007.

Jiang, Y., Luo, Y., Zhao, Z. and Tao, S.: Changes in wind speed over China during 1956–2004, Theor. Appl. Climatol., 99(3-4), 421–430, doi:10.1007/s00704-009-0152-7, 2009.

Jöhnk, K. D., Huisman, J., Sharples, J., Sommeijer, B., Visser, P. M. and Stroom, J. M.: Summer heatwaves promote blooms of harmful cyanobacteria, Glob. Change Biol., 14(3), 495–512, doi:10.1111/j.1365-2486.2007.01510.x, 2008.

Kamarainen, A. M., Penczykowski, R. M., Van de Bogert, M. C., Hanson, P. C. and Carpenter, S. R.: Phosphorus sources and demand during summer in a eutrophic lake, Aquat. Sci., 71(2), 214–227, doi:10.1007/s00027-009-9165-7, 2009.

Kara, E. L., Hanson, P., Hamilton, D., Hipsey, M. R., McMahon, K. D., Read, J. S., Winslow, L., Dedrick, J., Rose, K., Carey, C. C., Bertilsson, S., da Motta Marques, D., Beversdorf, L., Miller, T., Wu, C., Hsieh, Y.-F., Gaiser, E. and Kratz, T.: Time-scale dependence in numerical simulations: Assessment of physical, chemical, and biological predictions in a stratified lake at temporal scales of hours to months, Environ. Model. Softw., 35, 104–121, doi:10.1016/j.envsoft.2012.02.014, 2012.

King, J. R., Shuter, B. J. and Zimmerman, A. P.: The response of the thermal stratification of South Bay (Lake Huron) to climatic variability, Can. J. Fish. Aquat. Sci., 54(8), 1873–1882, doi:10.1139/f97-093, 1997.

[revised manuscript text omitted]

Shimoda, Y., Azim, M. E., Perhar, G., Ramin, M., Kenney, M. A., Sadraddini, S., Gudimov, A. and Arhonditsis, G. B.: Our current understanding of lake ecosystem response to climate change: What have we really learned from the north temperate deep lakes?, J. Gt. Lakes Res., 37(1), 173–193, doi:10.1016/j.jglr.2010.10.004, 2011.

Skowron, R.: Changeability of the ice cover on the lakes of northern Poland in the light of climatic changes, Bull. Geogr. Phys. Geogr. Ser., (1), 103–124, 2009.

Soranno, P. A., Carpenter, S. R. and Lathrop, R. C.: Internal phosphorus loading in Lake Mendota: response to external loads and weather, Can. J. Fish. Aquat. Sci. - CAN J Fish. AQUAT SCI, 54(8), 1883–1893, doi:10.1139/cjfas-54-8-1883, 1997.

Stauffer, R. E. and Armstrong, D. E.: Cycling of iron, manganese, silica, phosphorus, calcium and potassium in two stratified basins of Shagawa Lake, Minnesota, Geochim. Cosmochim. Acta, 50(2), 215–229, doi:10.1016/0016-7037(86)90171-7, 1986.

Stauffer, R. E. and Lee, G. F.: The role of thermocline migration in regulating algal blooms, in Modeling the eutrophication process, Ann Arbor Science Publishers, Inc, Ann Arbor, MI., 1973.

Stefan, H. G., Hondzo, M., Fang, X., Eaton, J. G. and McCormick, J. H.: Simulated long term temperature and dissolved oxygen characteristics of lakes in the north-central United States and associated fish habitat limits, Limnol. Oceanogr., 41(5), 1124–1135, doi:10.4319/lo.1996.41.5.1124, 1996.

Stenseth, N. C. and Mysterud, A.: Climate, changing phenology, and other life history traits: Nonlinearity and match–mismatch to the environment, Proc. Natl. Acad. Sci., 99(21), 13379–13381, doi:10.1073/pnas.212519399, 2002.

Stewart, K. M.: Physical limnology of some Madison lakes, PhD Thesis, University of Wisconsin-Madison, Madison, Wisconsin, USA., 1965.

Tanentzap, A. J., Hamilton, D. P. and Yan, N. D.: Calibrating the Dynamic Reservoir Simulation Model (DYRESM) and filling required data gaps for one-dimensional thermal profile predictions in a boreal lake, Limnol. Oceanogr. Methods, 5(12), 484–494, doi:10.4319/lom.2007.5.484, 2007.

Tanentzap, A. J., Yan, N. D., Keller, B., Girard, R., Heneberry, J., Gunn, J. M., Hamilton, D. P. and Taylor, P. A.: Cooling lakes while the world warms: Effects of forest regrowth and increased dissolved organic matter on the thermal regime of a temperate, urban lake, Limnol. Oceanogr., 53(1), 404–410, doi:10.4319/lo.2008.53.1.0404, 2008.

Tomé, A. R. and Miranda, P. M. A.: Piecewise linear fitting and trend changing points of climate parameters, Geophys. Res. Lett., 31(2), L02207, doi:10.1029/2003GL019100, 2004.

Toms, J. D. and Lesperance, M. L.: Piecewise regression: a tool for identifying ecological thresholds, Ecology, 84(8), 2034–2041, doi:10.1890/02-0472, 2003.

Van Cleave, K., Lenters, J. D., Wang, J. and Verhamme, E. M.: A regime shift in Lake Superior ice cover, evaporation, and water temperature following the warm El Niñ winter of 1997–1998, Limnol. Oceanogr., 59(6), 1889–1898, doi:10.4319/lo.2014.59.6.1889, 2014.

Vavrus, S. J., Wynne, R. H. and Foley, J. A.: Measuring the sensitivity of southern Wisconsin lake ice to climate variations and lake depth using a numerical model, Limnol. Oceanogr., 41(5), 822–831, doi:10.4319/lo.1996.41.5.0822, 1996.

Wan, H., Wang, X. L. and Swail, V. R.: Homogenization and trend analysis of Canadian near-surface wind speeds, J. Clim., 23(5), 1209–1225, doi:10.1175/2009JCLI3200.1, 2010.

Wilhelm, S. and Adrian, R.: Impact of summer warming on the thermal characteristics of a
polymictic lake and consequences for oxygen, nutrients and phytoplankton, Freshw. Biol.,
53(2), 226–237, doi:10.1111/j.1365-2427.2007.01887.x, 2008.

Williams, D. T., Drummond, G. R., Ford, D. E. and Robey, D. L.: Determination of light
extinction coefficients in lakes and reservoirs, in Proceedings of the Symposium on Surface
Water Impoundments, pp. 1329–1335, Minneapolis, MN, USA., 1980.

Williams, G., Layman, K. L. and Stefan, H. G.: Dependence of lake ice covers on climatic,
geographic and bathymetric variables, Cold Reg. Sci. Technol., 40(3), 145–164,
doi:10.1016/j.coldregions.2004.06.010, 2004.

Winslow, L. A., Read, J. S., Hansen, G. J. A. and Hanson, P. C.: Small lakes show muted
climate change signal in deepwater temperatures, Geophys. Res. Lett., 42(2), 2014GL062325,
doi:10.1002/2014GL062325, 2015.

Wrona, F. J., Prowse, T. D., Reist, J. D., Hobbie, J. E., Lévesque, L. M. J. and Vincent, W. F.:
Climate change effects on aquatic biota, ecosystem structure and function, Ambio, 35(7),
359–369, 2006.

Yan, N. D.: Effects of changes in pH on transparency and thermal regimes of Lohi Lake, near
Sudbury, Ontario, Can. J. Fish. Aquat. Sci., 40(5), 621–626, doi:10.1139/f83-081, 1983.

Yeates, P. S. and Imberger, J.: Pseudo two-dimensional simulations of internal and boundary
fluxes in stratified lakes and reservoirs, Int. J. River Basin Manag., 1(4), 297–319,
doi:10.1080/15715124.2003.9635214, 2003.

Ying, L., Shen, Z. and Piao, S.: The recent hiatus in global warming of the land surface:
Scale-dependent breakpoint occurrences in space and time: SCALE-DEPENDENT HIATUS
IN  SPACE  AND  TIME,  Geophys.  Res.  Lett.,  42(15),  6471–6478,
doi:10.1002/2015GL064884, 2015.

Table 1: Values of DYRESM-ICE parameters and model simulation specifications for both hydrodynamic and ice cover portions of the model.

| Hydrodynamic Model Parameters | Value |
| --- | --- |
| albedo | 0.08 [i,ii] |
| bulk aerodynamic momentum transport coefficient | 0.00139 [ii] |
| critical wind speed (m s$^{-1}$) | 4.3 [ii] |
| effective surface area coefficient (m$^2$) | $1 \times 10^7$ [iii] |
| emissivity of water surface | 0.96 [iv] |
| potential energy mixing efficiency | 0.2 [i,ii] |
| shear production efficiency | 0.06 [i, ii, iii] |
| vertical mixing coefficient | 200 [iii] |
| wind stirring efficiency | 0.8 [ii] |
| minimum layer thickness | 0.125 |
| maximum layer thickness | 0.6 [ii] |
| vertical light attenuation coefficient | variable [v] |

| Ice Model Parameters | Value |
| --- | --- |
| waveband 1, snow ice light extinction (m$^{-1}$) | 3.8 [vi] |
| waveband 2, snow ice light extinction( m$^{-1}$) | 20 [vi, vii] |
| waveband 1, blue ice light extinction (m$^{-1}$) | 1.5 [vi, vii] |
| waveband 2, blue ice light extinction (m$^{-1}$) | 20 [vi, vii] |
| waveband 1, snow light extinction (m$^{-1}$) | 6 [vi, vii] |
| waveband 2, snow light extinction (m$^{-1}$) | 20 [vi, vii] |
| distance of heat transfer, ice-water (m) | 0.039 [viii] |
| density, snow ice (kg m$^{-3}$) | 890 [vi] |
| density, blue ice (kg m$^{-3}$) | 917 [vi, vii] |
| density, snow (kg m$^{-3}$) | variable [ix] |
| compaction coefficient | variable [vi] |
| thermal conductivity, snow ice (W m$^{-1}$ °C$^{-1}$) | 2.0 [vi] |
| thermal conductivity, blue ice (W m$^{-1}$ °C$^{-1}$) | 2.3 [vi, vii] |
| thermal conductivity, snow (W m$^{-1}$ °C$^{-1}$) | variable [x] |
| thermal conductivity, sediment (W m$^{-1}$ °C$^{-1}$) | 1.2 [vi] |
| thermal conductivity, water (W m$^{-1}$ °C$^{-1}$) | 0.57 [vi, vii] |

sources: [i] Antenucci and Imerito, 2003; [ii]Tanentzap et al., 2007; [iii]Yeates and Imberger, 2003; [iv]Imberger and Patterson, 1981; [v]Williams et al., 1980; [vi]Rogers et al., 1995; [vii]Patterson and Hamblin, 1988; [viii]Vavrus et al., 1996; [ix]McKay, 1968; [x]Ashton, 1986

[revised manuscript text omitted]

0.193 (n = 104).

---

## Referee Comment (RC3) · Anonymous Referee #3 · 13 Apr 2016

This manuscript presents a study to demonstrate a long-term change in ice cover and thermal structure of Lake Mendota using a 1-D hydrodynamic-ice model. I have enjoyed reviewing this manuscript and I do believe that it is suitable for publication in a special issue of Hydrological Processes, in terms of its overall content. In general, the authors have made sound intellectual arguments and use an appropriate methodological approach, based on knowledge obtained from previous research studies. In addition, they provide a reasonable interpretation of the results obtained from this study.

General comments:

[Figure]

The author used daily meteorological data to run the DYRESM-WQ-I model, however is mentioned that the model has 1-hr time step at page 6 line 5!

The author used the rain fall and snow fall observations from weather station. Snow accumulation regimes differ significantly not only between but also within the various locations over a lake. Snow depth can be very thin and dense to non-existent on some lakes or lake sections due to the wind. This difficulty in accurately measuring snowfall have to be considered specially when running 1-D models. This can be done by looking at any available in-situ snow observations over lake and calculating the percentage of snow in comparison with the station data. And also it is not clear how the snow density is defined. Is there any in-situ observations available for snow density?

Some specific questions/comments I have about this manuscript are as follows:

Page 2/lines 24-25: "Air temperature, wind speed, and water clarity are important factors driving these lake ecosystem properties", reference is missing.

Page 2/lines 27-29: "The long-term response of lake ice and water temperature to changing air temperature and wind speed is integral to assessment of the potential impacts of climate change on water quality and ecology of lakes." It is not clear if the author is talking about the role of lake ice on climate or the response of the lakes on climate. Please be more specific.

Page 8, 3.2 meteorological variables: the location where the meteorological data are collected is not clear and how far it is in comparison with the simulation points.

Page 12/lines 22-26: "Other models including LIMNOS (Vavrus et al. 1996) on Lake Mendota, Wisconsin; MLI (Rogers et al. 1995) on Harmon Lake, British Columbia; and CLIMo (Duguay et al. 2003) on lakes in Barrow, Alaska; Poker Flat, Alaska; and Churchill, Manitoba produced similar errors to Lake Mendota between modeled and observed ice thickness and snow cover". This comparison should be more specific and the author have to give a range of error.

Please also note the supplement to this comment:
http://www.hydrol-earth-syst-sci-discuss.net/hess-2015-488/hess-2015-488-RC3-supplement.pdf

―――――――――――――――――――――

---

## Author Comment (AC5) · 19 Apr 2016

The authors would like to thank Dr. Homa Kheyrollah Pour for providing the valuable comments to further improve the quality of the manuscript. We address the comments and detailed these changes in the point-by-point response below.

This manuscript presents a study to demonstrate a long-term change in ice cover and thermal structure of Lake Mendota using a 1-D hydrodynamic-ice model. I have enjoyed reviewing this manuscript and I do believe that it is suitable for publication in a special issue of Hydrological Processes, in terms of its overall content. In general, the

authors have made sound intellectual arguments and use an appropriate methodological approach, based on knowledge obtained from previous research studies. In addition, they provide a reasonable interpretation of the results obtained from this study. We thank the reviewer for the compliments.

General comments: The author used daily meteorological data to run the DYRESM-WQ-I model, however is mentioned that the model has 1-hr time step at page 6 line 5!

Yes, this is correct. We used a daily timestep for meteorological and inflow/outflow inputs based on the availability of data to complete the study (subdaily data availability was not consistently available for the 100+ year time period). Model output was also at daily intervals. However, the model itself performs calculations at a 1 hour timestep both for the hydrodynamic and ice cover components of the model. This 1 hour timestep was chosen to ensure that change in ice depth is relatively small because the time step is small.

The author used the rain fall and snow fall observations from weather station. Snow accumulation regimes differ significantly not only between but also within the various locations over a lake. Snow depth can be very thin and dense to non-existent on some lakes or lake sections due to the wind. This difficulty in accurately measuring snowfall has to be considered specially when running 1-D models. This can be done by looking at any available in-situ snow observations over lake and calculating the percentage of snow in comparison with the station data. And also it is not clear how the snow density is defined. Are there any in-situ observations available for snow density?

We thank the reviewer for this valuable suggestion. We acknowledge that the snow depth is variable both between the meteorological station and the lake and across the lake itself. In-situ snow observations over the lake are generally limited to one location once per year for only a limited subset of the 104 year model period. During the winter 2009-2010, when ice and snow measurements were taken multiple times, snow depth

matches well with errors of <1 cm to 7 cm during the year (see Figure 3a). As we only have 49 snow measurements for 104 years of simulation data, adjusting snow amounts by first comparing in-situ observations to station data may not be appropriate. However, we will definitely use this approach to improve model ability during shorter-term studies in the future. Snow density is predicted according to Equation 11 in Rogers et al (1995 - see manuscript reference list). $s = n + (m - n)\{1 - [\exp(-kd)]\}$. $k = -\ln[(m - l)/(m - n)]$, $n$ is the density of fresh snow, $l$ the density 24 h after falling, $m$ the maximum density, and $d$ the number of days since the last snowfall. To our knowledge there are no in-situ measurements of snow density for Lake Mendota.

Some specific questions/comments I have about this manuscript are as follows:

Page 2/lines 24-25: "Air temperature, wind speed, and water clarity are important factors driving these lake ecosystem properties", reference is missing.

The line has been updated as follows to add references: "Air temperature (Findlay et al., 2001; Lynch et al., 2015), wind speed (Brown et al., 1993; Lynch et al., 2015), and water clarity (Arhonditsis et al., 2004b; Lathrop et al., 1996)."

Page 2/lines 27-29: "The long-term response of lake ice and water temperature to changing air temperature and wind speed is integral to assessment of the potential impacts of climate change on water quality and ecology of lakes." It is not clear if the author is talking about the role of lake ice on climate or the response of the lakes on climate. Please be more specific.

We thank the reviewer for pointing out this unclear sentence. The authors are discussing the response of lake ice and water temperature to changes in climate and the corresponding impact of lake ice and water temperature changes on lake ecology. The sentence has been edited for clarity as follows: "The response of lake ice and water temperature to long-term changes in air temperature and wind speed is integral to assess potential impacts of climate change on lake ecology."

Page 8, 3.2 meteorological variables: the location where the meteorological data are collected is not clear and how far it is in comparison with the simulation points.

All data except for solar radiation is obtained from the weather station of the National Climate Data Center (NCDC, NOASS) locate in Madison (MSN) Dane County Regional Airport (Truax Field), which is approximately 4.5 km east of the simulation location. Solar radiation data are obtained from the St. Charles, Illinois weather station, approximately 150 km southeast of Lake Mendota. Section 3.2 has been updated to clarify where the meteorological data were obtained and how far those locations are from the study site, Lake Mendota. Additions are provided in the following.

"Meteorological data for the Madison area have been continuously recorded since 1869, however, the station and techniques have changed several times. Robertson (1989) constructed a continuous, homogeneous daily meteorological dataset from 1884 to 1988 by adjusting for changes in site location and observation time, and resultant changes in the surface roughness (e.g. height of surrounding trees and buildings). These data were appended with data from the most recent weather station of the National Climate Data Center (NCDC, NOAA) located in Madison (MSN) Dane County Regional Airport (Truax Field), approximately 4.5 km east from the simulation location, the same site as that used in 1988. All data except solar radiation can be obtained for Madison (MSN) from http://www.ncdc.noaa.gov/, except solar radiation which can be obtained from http://www.sws.uiuc.edu/warm/weather/ (approximately 150 km southeast of Lake Mendota). Since Robertson (1989) adjusted all historical data to that collected in 1988, no adjustments are applied to the recent data except for wind. In 1996, a discontinuity in the wind record was caused by change in observational techniques and sensor locations (McKee et al., 2000). To address the non-climatic changes in wind speed, data from MSN are carefully compared with those collected from the tower of the Atmospheric and Oceanic Science Building at the University of Wisconsin-Madison (http://ginsea.aos.wisc.edu/labs/mendota/index.htm), approximately 4 km south of simulation location. Hourly data from both sites (UMSN,hourly and UAOS,hourly) during

2003–2010 were used to form a 4×12 (four components of wind direction × 12 months) matrix (K4,12) of wind correction factors, yielding UAOS,daily= Ki,j×UMSN,daily. A comparison of results indicated that the MSN weather station measured a higher magnitude in winds out of the east by 5% and lower magnitude in winds out of the west and south by 30% and 10%, respectively. The adjusted wind data (=Ki,j×UMSN,daily) are used in the model simulation. Overall the adjusted wind data show a decline in mean wind velocities of 16% from 1988−93 to 1994−2014) compared to 7% at a nearby weather station with no known observational changes (St. Charles, Illinois; 150 km southeast of Lake Mendota)."

Page 12/lines 22-26: "Other models including LIMNOS (Vavrus et al. 1996) on Lake Mendota, Wisconsin; MLI (Rogers et al. 1995) on Harmon Lake, British Columbia; and CLIMo (Duguay et al. 2003) on lakes in Barrow, Alaska; Poker Flat, Alaska; and Churchill, Manitoba produced similar errors to Lake Mendota between modeled and observed ice thickness and snow cover". This comparison should be more specific and the authors have to give a range of error.

To address comment, we have revised the writing as follows (see Page 13/line 20-30): "In comparison, similar discrepancies between modelled and observed ice thickness and snow cover were produced from other models including LIMNOS (Vavrus et al., 1996) on Lake Mendota, Wisconsin with discrepancies of 4−9 cm for ice cover; MLI (Rogers et al., 1995) on Harmon Lake, British Columbia, which had up to 6 cm error for ice cover and 4 cm error for snow cover; and CLIMo (Duguay et al., 2003) on lakes in Barrow, Alaska (differences of 5−6 cm for ice thickness); Poker Flat, Alaska (mean absolute error of 2 cm for ice cover and underestimation of snow - ice thickness of 7 cm); and Churchill, Manitoba (Ice thickness observations were within model values for the snow-free and 100% snow covered scenarios). Duguay et al. (2003) found that variability in snow density and snow accumulation play a significant role in ice thickness, which may account for discrepancies between simulated and observed ice cover thicknesses in our study

Please also note the supplement to this comment:
http://www.hydrol-earth-syst-sci-discuss.net/hess-2015-488/hess-2015-488-AC5-supplement.pdf

―――――――――――――――――